# Antarctic Tipping points triggered by the mid-Pliocene warm climate

Javier Blasco[1], Ilaria Tabone[2], Daniel Moreno-Parada[3,4], Alexander Robinson[5], Jorge Alvarez-Solas[4], Frank Pattyn[1], and Marisa Montoya[3,4]

[1]Laboratoire de Glaciologie, Université libre de Bruxelles, Brussels, Belgium
[2]Friedrich Alexander Universität Erlangen-Nürnberg, Institut für Geographie, Erlangen, Germany
[3]Departamento de Física de la Tierra y Astrofísica, Universidad Complutense de Madrid, Facultad de Ciencias Físicas, Madrid, Spain
[4]Instituto de Geociencias, Consejo Superior de Investigaciones Científícas-Universidad Complutense de Madrid, Madrid, Spain
[5]Alfred Wegener Institute, Helmholtz Centre for Polar and Marine Research, Potsdam, Germany

**Correspondence:** J. Blasco (javier.blasco.navarro@ulb.be)

**Abstract.**

Tipping elements, including the Antarctic Ice Sheet (AIS), are Earth system components that could reach critical thresholds due to anthropogenic emissions. Increasing our understanding of past warm climates can help to elucidate the future contribution of the AIS to emissions. The mid-Pliocene warm period (mPWP, $\sim$3.3-3.0 million years ago) serves as an ideal benchmark experiment. During this period, $CO_2$ levels were similar to present-day (PD, 350-450 ppmv), but global mean temperatures were 2.5-4.0 K higher. Sea-level reconstructions from that time indicate a rise of 5-25 meters compared to the present, highlighting the potential crossing of tipping points in Antarctica. In order to achieve a sea-level contribution far beyond 10 m not only the West Antarctic Ice Sheet (WAIS) needs to largely decrease, but a significant response in the East Antarctic Ice Sheet (EAIS) is also required. A key question in reconstructions and simulations is therefore which of the AIS basins retreated during the mPWP. In this study, we investigate how the AIS responds to climatic and bedrock conditions during the mPWP. To this end we use the Pliocene Model Intercomparison Project, Phase 2 (PlioMIP2) General Circulation Model ensemble to force a higher-order ice-sheet model. Our simulations reveal that the WAIS experiences collapse with a 0.5 K oceanic warming. The Wilkes basin shows retreat at 3 K oceanic warming, although higher precipitation rates could mitigate such a retreat. Totten glacier shows slight signs of retreats only under high oceanic warming conditions (greater than 4 K oceanic anomaly). If only the WAIS collapses, we simulate a mean contribution of $2.7^{+0.1}_{-0.4}$ mSLE to $7.0^{+0.1}_{-0.1}$ mSLE (uncertainty represents the interquartile range). If, in addition, the Wilkes basin retreats, our simulations suggest a mean contribution of $6.0^{+1.8}_{-1.3}$ mSLE to $8.9^{+0.2}_{-0.3}$ mSLE. Besides uncertainties related to the climate forcing, we also examine other sources of uncertainty related to initial ice thickness and ice dynamics. We find that the climatologies yield a higher uncertainty than the dynamical configuration, if parameters are constrained with PD observations, and that starting from Pliocene reconstructions leads to smaller ice-sheet configurations due to the hysteresis behaviour of marine bedrocks. Ultimately, our study concludes that marine ice cliff instability is not a prerequisite for the retreat of the Wilkes basin. Instead, a significant rise in oceanic temperatures can initiate such a retreat.

# 1 Introduction

Sea level has been rising since the beginning of the 20th century due to thermal expansion and melting of glaciers and ice sheets (Frederikse et al., 2020). Sea level will continue to rise by the end of this century and very likely far beyond that period depending on the followed future emission pathways (IPCC AR6; Masson-Delmotte et al. (2021)). The Antarctic Ice Sheet (AIS) plays a major role in future sea-level projections, as it is the largest ice sheet on Earth, with a total volume of ~58 meters of sea-level equivalent (mSLE; Morlighem et al. (2020)). Nonetheless, assessment of its future contribution using ice-sheet models is subject to a very large uncertainty, mainly due to our poor understanding of ice-sheet-related physical processes (Seroussi et al., 2020; van de Wal et al., 2022). From a tipping point perspective, modeling studies suggest that the AIS exhibits three potential critical thresholds (Armstrong McKay et al., 2022): a collapse of the West Antarctic Ice Sheet (WAIS), which is likely to occur below 2 K of warming since the pre-industrial era (Sutter et al., 2016; Garbe et al., 2020); a collapse of the marine basins of the East Antarctic Ice Sheet (EAIS), with a tipping point between 2-4 K (Garbe et al., 2020; DeConto et al., 2021); and a fully melted EAIS, probably above 8 K of global warming (Garbe et al., 2020). In this study we will mainly focus on the internal feedback mechanisms that can lead to a collapse of the marine basins in the WAIS and EAIS.

The WAIS, as well as many regions of the EAIS, lies on marine bedrock with a retrograde slope. If the grounding line retreats into pronounced bed-slopes, the Marine Ice Sheet Instability (MISI; Schoof (2007)) can be initiated. Such a retreat can be triggered by the shrinking of ice shelves. Although ice shelves do not directly contribute to sea-level rise, they can help reduce inland ice velocities due to their buttressing effect (Fürst et al., 2016; Sun et al., 2020). The stability of ice shelves depends on several processes, such as increased oceanic melt (Rignot et al., 2013), hydrofracturing (Robel and Banwell, 2019) or ice damage (Lhermitte et al., 2020). Therefore, one key question regarding the AIS stability is whether there is a temperature threshold at which AIS ice shelves are not large enough to provide the necessary buttressing effect to the interior of the ice sheet, triggering MISI and eventually leading to a collapse of its marine regions.

Sea-level reconstructions suggest that AIS marine regions indeed collapsed during past warmer periods, highlighting the importance of the assessment of AIS tipping points (Rohling et al., 2014, 2019). One of these warmer periods is the mid-Pliocene Warm Period (~3.3-3.0 Ma). This period was characterized by atmospheric $CO_2$ concentrations similar to present day (PD) values (350-450 ppmv), although with significantly warmer global temperatures (2.5-4 K; Haywood et al. 2016; De La Vega et al. 2020; Guillermic et al. 2022) which could reach up to 8 K at high latitudes due to polar amplification (Fischer et al., 2018). Sea-level reconstructions of that period show high uncertainty, yet they suggest that sea level was considerably higher than today. The highest global estimated sea-level contribution during the Pliocene comes from Hearty et al. (2020), with reconstructions in the South African coast far above 30 meters of sea-level rise. Dumitru et al. (2019) reconstruct a total of 25 meters of sea-level rise (23.4 mSLE from ice sheets and 1.6 meters from thermal expansion) from caves in Mallorca. Model results, on the other hand, lower these sea-level contributions to 8-20 meters of sea-level rise (Moucha and Ruetenik, 2017; Grant et al., 2019; Richards et al., 2022). Such high sea-level stands point to a substantial contribution of continental ice sheets. Even if the Greenland Ice Sheet was entirely absent (which holds around 7 mSLE), it is still necessary to account for a

substantial Antarctic contribution to achieve the reconstructed Pliocene sea level. Thus, it is very likely that Antarctic tipping points were exceeded during the mPWP, making this an ideal benchmark period for assessing AIS stability in warmer climates.

Ice-sheet modeling studies also suggest a wide range of AIS contributions to sea-level rise during the mPWP. Dolan et al. (2018) forced three ice-sheet models with climate output from seven Atmosphere-Ocean General Circulation Models (AOGCMs) in the frame of the first stage of the Pliocene Model Intercomparison Project (PlioMIP1). They showed that, although climatologies can lead to important differences, the largest source of uncertainty is the ice-sheet model used, stressing the importance of analyzing the sources of structural uncertainty. Golledge et al. (2017) simulated two Antarctic states (by allowing in one case for melting at the grounding line) and performed an analysis with varying climatic conditions. They found a mean AIS contribution of 8.6 mSLE (9.7 mSLE if melting at the grounding line is allowed). Yan et al. (2016) investigated Antarctic sea-level uncertainty in their ice-sheet model to model parameters and climatic sensitivities. They found a mean Antarctic contribution of 5.6 mSLE but parameter uncertainty in their model ensemble shows a spread of 10.8 mSLE which led even to negative sea-level contributions. Finally, Berends et al. (2019) simulated a total sea-level rise of 8–14 mSLE during the late Pliocene accounting for the contribution from all ice sheets. The largest simulated Antarctic sea-level contributions at the mPWP are provided by the studies of DeConto and Pollard (2016) and DeConto et al. (2021), with simulated means of 11.3 mSLE and 17.8 mSLE, respectively. In both cases, they performed a large ensemble analysis testing parameters that affect ice-shelf sensitivity, such as maximum calving and the hydrofracturing rate on ice shelves. In these studies, the large contribution is due to the inclusion of the so-called Marine Ice Cliff Instability (MICI), a potential positive feedback mechanism that affects marine terminating glaciers. Marine cliffs that form at the ice front are thought to fail when their thickness exceeds a certain threshold. The retreat rate of marine cliffs increases with ice thickness (Crawford et al., 2021). Thus, if an ice front retreats and encounters a higher ice thickness upstream, the retreat rate increases, accelerating the grounding-line flux. Although the physics of such a mechanism are becoming more clear thanks to idealized experiments (Bassis et al., 2021; Crawford et al., 2021), its application to the AIS remains a matter of debate (Edwards et al., 2019).

Another approach to infer sea-level estimates from a modelling perspective is through Geodynamic Models. These models use glacial isostatic adjustment and mantle dynamic topography to compute sea-level estimates, distinguishing regional and global sea-level increase. An advantage is that they account for potential rebound effects which are difficult to assess from in-situ sea-level records. Hollyday et al. (2023) used such a model to simulate the mantle flow from the Patagonian region. This allowed them to lower mPWP sea-level estimates to 17.5±6.4 mSLE, and to assess the AIS contribution in 9.5±6.9 mSLE. Similar results are obtained by Richards et al. (2022) by simulating the Australian mantle deformation and comparing it with proxy data from that region. They obtain a mPWP sea-level stand from 10.4-21.5 mSLE. Moucha and Ruetenik (2017) simulate a global sea-level contribution of 15 mSLE based on the US Atlantic shoreline. These studies reflect an overestimation in sea-level rise of in-situ records since lithospheric rebound is poorly considered.

One key question in Antarctic reconstructions and simulations is whether the Wilkes Basin retreated or not during the mPWP (Wilkes basin illustrated in Fig. S1). Today, the WAIS and the Greenland Ice Sheet (GrIS) sum up to make a total of 10 mSLE (Morlighem et al., 2017, 2020). Thus, in order to achieve a sea-level rise far beyond 10 mSLE, a significant response of the EAIS is required. Marine records close to the Wilkes basin reinforce the hypothesis of such a retreat. Deposition of ice-rafted

debris show enhanced iceberg activity during the mPWP (Patterson et al., 2014; Bertram et al., 2018). This can be interpreted as a consequence of ice-sheet retreat with its consequent calving events. In addition, land-based sediment records of the EAIS show low concentrations of cosmogenic isotopes, which indicates that land-based regions experienced minimal retreat during the mPWP (Shakun et al., 2018). This points to a response of marine-based regions to explain the high sea-level records.

From an ice-sheet modeling perspective, DeConto and Pollard (2016) and DeConto et al. (2021) achieved the most retreated EAIS, especially in the Wilkes Basin, due to the inclusion of the MICI mechanism. Golledge et al. (2017) obtained a collapse of the Wilkes basin by warming the Pliocene climate by 2 K in the atmosphere and 1 K in the ocean. Yan et al. (2016) also achieved a collapsed Wilkes Basin, but only for an additional 5 K oceanic warming. Dolan et al. (2018) and de Boer et al. (2014) simulated a collapsed Wilkes basin when the model was initialized with boundary conditions of the third phase of the Pliocene Research, Interpretation and Synoptic Mapping (PRISM3), which included higher $CO_2$ concentrations than today and a different paleo ice-sheet geography and topography. In the transient simulation of Berends et al. (2019) only a WAIS collapse is achieved.

Our purpose here is to explore the AIS contribution to sea-level rise during the mPWP and to assess potential tipping points that can lead from a PD configuration to a mPWP state. Here we present the response of the Yelmo ice-sheet-shelf model to the mPWP climate simulated during Phase 2 of the PlioMIP project. The aim is to investigate parameter uncertainties of the ice-sheet model and their impact on the resulting simulations, as well as climatological uncertainties from the PlioMIP2 AOGCMs. The study is structured as follows: first we describe the ice-sheet model and the experimental setup (Section 2). Then, the main results of the PlioMIP2-forced experiments are shown (Section 3). Our results are compared with those from other ice-sheet models and reconstructions. A discussion of our simulations (Section 4) is followed by the main conclusions (Section 5).

## 2  Methods and experimental setup

### 2.1  Yelmo ice-sheet-shelf model

For this study we use the Yelmo ice-sheet-shelf model with a horizontal resolution of 16 km with 21 sigma coordinates in the vertical dimension. Yelmo is thermomechanically coupled and uses Glen's flow law with an exponent of n=3. Ice velocities are computed via the depth-integrated-viscosity approximation (DIVA; Goldberg (2011)). The DIVA solver replaces the horizontal velocity gradients and effective viscosity by their vertical averages, which makes it computationally efficient, but still allows obtaining results similar to other 3D higher-order models (Robinson et al., 2022). Here we will describe the most important features used in our experimental setup. Additional information on Yelmo is provided by Robinson et al. (2020).

#### Basal-drag law

Basal friction at the ice bed is represented with a regularized-Coulomb friction law

$$\boldsymbol{\tau}_{\mathrm{b}} = c_{\mathrm{b}} \left( \frac{|\mathbf{u}_{\mathrm{b}}|}{|\mathbf{u}_{\mathrm{b}}| + u_0} \right)^q \frac{\mathbf{u}_{\mathrm{b}}}{|\mathbf{u}_{\mathrm{b}}|}, \tag{1}$$

with basal velocity $u_{\mathrm{b}}$. The regularization constant $u_0$ is set to $100 \ \mathrm{m/yr}$ following Zoet and Iverson (2020), while $q$ is the friction law exponent that determines the ice flow regime. The spatially variable basal friction coefficient $c_{\mathrm{b}}$ is defined as

$$c_{\mathrm{b}} = c_{\mathrm{f}} \lambda N, \ \text{with} \tag{2}$$

$$\lambda = \begin{cases} 1 & \text{if } z_b \geq 0 \\ \max \left[ \exp \left( -\frac{|z_b|}{z_0} \right), 10^{-4} \right] & \text{if } z_b < 0 \end{cases}. \tag{3}$$

Here, $c_{\mathrm{f}}$ is a dimensionless field representing the basal properties of the base, such as soft ($c_{\mathrm{f}} = 0.1$) or hard beds ($c_{\mathrm{f}} = 1.0$). Here we will use it for tuning of the model as described in the spin-up procedure. $N$ is the effective pressure dependent on the overburden pressure as in the formulation of Leguy et al. (2014). $\lambda$ is a scaling factor which follows an exponential dependency with the bedrock height ($z_b$) with an e-folding depth of $z_0 = 400 \ \mathrm{meters}$ following Blasco et al. (2021). This ensures that ice flows faster in marine regions due to softer soil properties. All parameter values are summarized in Table 1.

**Grounding-line treatment**

The grounding line is defined via the flotation criterion. In order to trace its position accurately in transient experiments it is necessary to use high resolution in its vicinity (Pattyn et al., 2013). However, this leads to a high computational cost and hinders studies that involve long timescales, such as transient paleoclimatic studies. In order to overcome this problem, basal friction is scaled at the grounding-line points, where floating and grounded ice coexist, with the grounded fraction of the grid cell. This method has shown to lead to good results for coarse resolution (Feldmann et al., 2014; Leguy et al., 2021; Berends et al., 2022) and to convergence in Yelmo with higher resolution (Robinson et al., 2020). We do not apply melting at the grounding line to avoid overestimation of the ocean-induced retreat in our simulations (Seroussi and Morlighem, 2018).

**Calving**

The calving rate $C$ is derived as a sum between the principal stresses ($\tau_1$ and $\tau_2$) as in Lipscomb et al. (2019):

$$C = \kappa_{\mathrm{t}} \tau_{\mathrm{ec}}, \ \text{with} \tag{4}$$

$$\tau_{\mathrm{ec}}^2 = \max (\tau_1, 0)^2 + \omega_2 \max (\tau_2, 0)^2, \tag{5}$$

where $\kappa_{\mathrm{t}}$ and $\omega_2$ are constants used to mimic ice extent as close as possible to observations (see Table 1).

**Boundary conditions**

If the temperature at the ice base reaches the pressure melting point, it remains at that value and the basal mass balance is diagnosed as in Cuffey and Paterson (2010), where the geothermal heat flow field is obtained from Davies (2013). The glacial isostatic adjustment is computed with the elastic lithosphere-relaxed asthenosphere method (Le Meur and Huybrechts, 1996), where the relaxation time of the asthenosphere is set to 3000 years.

## 2.2 Climatic forcing

**Surface mass balance**

Surface melt is computed via the Insolation-temperature melt (ITM) method (Pellicciotti et al., 2005; Van Den Berg et al., 2008; Robinson et al., 2010). Daily surface melt is obtained from surface air temperature and absorbed insolation:

$$M_{\mathrm{srf}} = \frac{\Delta t}{\rho_{\mathrm{w}} L_{\mathrm{i}}} \left[ \tau_{\mathrm{a}} \left( 1 - \alpha_{\mathrm{s}} \right) S + c + \lambda_{\mathrm{srf}} T_{\mathrm{srf}} \right], \tag{6}$$

where $\tau_{\mathrm{a}}$ is the transmissivity of the atmosphere (i.e., the ratio between downward shortwave radiation at the land surface
and at the top of the atmosphere), $\rho_{\mathrm{w}}$ the density of pure water, $L_{\mathrm{i}}$ is the latent heat of ice, $\alpha_{\mathrm{s}}$ the surface albedo of snow, $S$ the insolation at the top of the atmosphere and $\Delta t$ the day length in seconds. $\lambda_{\mathrm{srf}}$ and $c$ are parameters used to calibrate the AIS ice thickness and extension (Table 1). This method accounts for the shortwave radiation and differences between snow and ice through the albedo effect. From the total computed melting we assume that a 60% refreezes again as in Robinson et al. (2010).

**Atmospheric forcing**

Ice-sheet surface atmospheric temperatures and precipitation rates are obtained either from reanalysis or from climatic models. In order to investigate the response of the AIS to the mPWP climate we use an anomaly method similar to Blasco et al. (2021):

$$T_{\mathrm{mPWP}}^{\mathrm{atm}} = T_{\mathrm{pd}}^{\mathrm{atm}} + \Delta T_{\mathrm{mPWP}}^{\mathrm{atm}} \tag{7}$$

$$P_{\mathrm{mPWP}} = P_{\mathrm{pd}} \cdot \delta P_{\mathrm{mPWP}}. \tag{8}$$

Here the subindex $_{\mathrm{pd}}$ stands for present-day climate. These fields are obtained from the regional atmospheric climate model
RACMO2.3 (Van Wessem et al., 2014) forced with the ERA-Interim reanalysis data (Dee et al., 2011). The temperature anomaly fields ($\Delta T_{\mathrm{mPWP}}^{\mathrm{atm}}$) and relative precipitation fields ($\delta P_{\mathrm{mPWP}}$) are computed between the Pliocene experiment with 400 ppmv $CO_2$ and the pre-industrial control run using 280 ppmv $CO_2$ from 12 different AOGCMs in the frame of PlioMIP2 (see Haywood et al. (2016) for more information on the experimental setup). In order to account for surface temperature and precipitation changes in elevation a lapse rate correction factor $\Gamma$ is applied, which is equal to 0.008 K m$^{-1}$ for annual
temperatures and to 0.0065 K m$^{-1}$ for summer temperatures (Ritz et al., 1996; DeConto and Pollard, 2016; Quiquet et al., 2018; Albrecht et al., 2020)

$$T_{\text{mPWP}}^{\text{atm}}\left(z_{\text{srf}}\right) = T_{\text{mPWP}}^{\text{atm}} - \Gamma \, z_{\text{srf}} \tag{9}$$

$$P_{\text{mPWP}}\left(z_{\text{srf}}\right) = P_{\text{mPWP}} \, \exp\left(-f \, \Gamma \, z_{\text{srf}}\right), \tag{10}$$

where $f$ is a precipitation change factor set to $0.05 \; ^{\circ}\text{K}^{-1}$ in the Clausius-Clapeyron relation. Note that in order to compute
these fields, we needed to scale the PD and mPWP climatologies to PD sea-level elevation using the same equations. PD climatologies are scaled with the surface elevation from RACMO2.3, whereas the mPWP climatologies are scaled with the surface elevations provided by the PlioMIP2 protocol. This ensures that surface changes, as well as any elevation bias from RACMO2.3 or PlioMIP2 fields are taken into account in our model. Figures 1 and 2 show the anomaly fields from the 12 AOGCMs used in this study scaled to sea-level elevation.

**Ocean forcing**

Here we use a quadratic local and non-local law following a similar approach to that of the ISMIP6 protocol (Jourdain et al., 2020). The quadratic local and non-local law not only includes local temperature changes, but also the average over the ice-shelf basin. This parameterisation accounts for the overturning circulation below the ice-shelf cavities, which affects the total basal melt in a non-linear way (Favier et al., 2019). It reads as follows:

$$M_{\text{quad-nl}} = \gamma_{\text{quad-nl}} \left(\frac{\rho_{\text{sw}} c_{\text{po}}}{\rho_{\text{i}} L_{\text{i}}}\right)^2 \langle T^{\text{ocn}} - T_{\text{f}} \rangle |T^{\text{ocn}} - T_{\text{f}}|, \tag{11}$$

where $\gamma_{\text{quad-nl}}$ represents the heat exchange velocity, $\rho_{\text{sw}}$ and $\rho_{\text{i}}$ the ocean water and ice densities, respectively, $c_{\text{po}}$ the specific heat capacity of the ocean mixed layer, and $L_{\text{i}}$ the latent heat of fusion of ice (Table 1). The freezing point temperature $T_{\text{f}}$ at the ice-shelf base is defined as:

$$T_{\text{f}} = \lambda_1 S^{\text{ocn}} + \lambda_2 + \lambda_3 z_{\text{b}}, \tag{12}$$

where $z_{\text{b}}$ represents the ice-base elevation (negative below sea level), and the coefficients $\lambda_1$, $\lambda_2$, and $\lambda_3$ are respectively the liquidus slope, intercept, and pressure coefficient (Table 1). Ocean temperature and salinity ($T_{\text{f}}$ and $S^{\text{ocn}}$, respectively) are three-dimensional oceanic fields. PD fields are obtained from the World Ocean Dataset and extrapolated into the sub-shelf cavities following the ISMIP6 protocol (Jourdain et al., 2020). mPWP fields are obtained from the AOGCMs outputs. Four of the 12 PlioMIP2 climate simulations, however, did not provide oceanic data. For those cases, a spatially homogeneous
temperature anomaly field of one fourth of the atmospheric anomaly was applied, following work by Golledge et al. (2015) and Taylor et al. (2012). The resulting thermal forcing anomaly field between the mPWP and PD is shown in Figure 3.

Note that the AOGCMs did not provide any oceanic information for Antarctic grid cells. Since we need that grid information to force our ice-sheet model, we interpolated to that grid point using the value of the nearest neighbor at the same depth. Of course, applying other interpolation schemes - and increasing the spatial resolution of the grid - could potentially change the

oceanic conditions and lead to slightly different final states. Nonetheless, since our aim here was to assess tipping points of the AIS, we decided to stay with the nearest neighbor interpolation for simplicity.

## 2.3 Experimental setup

### Present-day spin up

First we perform an ensemble of 150 ice-sheet simulations for the AIS with different dynamic configurations under steady PD climatic conditions using the PD forcing fields. The ice-sheet dynamics, thermodynamics and topography are allowed to evolve freely. This approach differs from other studies, where friction coefficients are optimized to simulate an AIS as close as possible to observations. Instead we use the more general friction coefficients that vary depending on the bedrock properties as described above since there is no a priori reason to believe that optimized friction coefficients for PD would have been the same for the mPWP. However, we do a test of experiments with optimized friction coefficient fields, which we mention in the Discussions section.

We investigate uncertainty arising from three parameters that affect the ice dynamics: the exponent of the friction law $q$, the enhancement factor $E_{\mathrm{f}}$ and the friction coefficient $c_{\mathrm{f}}$ (Table 1). The friction exponent and coefficient affect the basal friction directly. Ten values are chosen for the friction coefficient, from 0.1 to 1.0 in steps of 0.1. Three values are chosen for the friction exponent: 0.0, 0.2, 1.0. The enhancement factor is a typical arbitrary scalar introduced in the Arrhenius equation to approximate the effect of an anisotropic flow. It is chosen from 1 to 5 in steps of 1 following values explored in Ma et al. (2010) (Table 1). The simulations are run for 100 kyr to ensure equilibration at the PD. Simulations are considered realistic if the simulated PD ice-volume differs by less than 1 mSLE and grounded-ice differs by less than 2% from observations from Morlighem et al. (2020). With these criteria, we simulate a PD state comparable to other ice-sheet models (Seroussi et al., 2019). From our 150 ensemble members, only 31 simulations fulfill these conditions (Fig. S2, S3, S4).

### Paleo simulations

The 31 selected model versions are then used to simulate mPWP conditions with forcing from the 12 different AOGCMs. This gives a total of 372 simulations. These simulations are initialized from the end of the respective PD simulation and forced under steady mPWP conditions until they reach a new equilibrated state; after 30 kyr no significant changes are observed either in ice volume or in ice area, (Figure S2, S3, S4). The background global sea level is set to 20 meters above PD for all simulations, representative of the highest estimates. Assuming a fixed and stable mPWP climatic state is a simplification compared to reality, since the AIS ice volume and climate vary through time (Yan et al., 2016; DeConto and Pollard, 2016; Golledge et al., 2017). However, this approach allows us to make use of the PlioMIP AOGCM ensemble and to perform a straightforward comparison to gain insight into model sensitivities to climatic forcing.

## 3 Results

### 3.1 Ensemble simulations

The simulated ice volumes (in meters of sea-level equivalent, mSLE) and ice extent at equilibrium are shown in Figures 4 and 5. All AOGCMs show a smaller AIS in terms of volume and extension with one exception (MIROC4m). Based on sea-level reconstructions, MIROC4m cannot be considered as realistic, nonetheless, we will discuss the potential reason for this unexpected behavior in the following sections. Over the remaining simulations, the simulated sea-level contributions range from $2.7^{+0.1}_{-0.4}$ mSLE (HadGEM3; the uncertainty represents the interquartile range) to $8.9^{+0.2}_{-0.3}$ mSLE (EC-Earth3.3). Ice extent ranges from $9.25^{+0.02}_{-0.03}$ $10^6$ km$^2$ (EC-Earth3.3) to $10.77^{+0.03}_{-0.05}$ $10^6$ km$^2$ (NorESM1-F). For reference, the PD grounded extension lies around 12.3 x $10^6$ km$^2$, while an extent of around 10 x $10^6$ km$^2$ represents a collapsed WAIS basin and even lower numbers indicate a retreat of marine basins in the EAIS. Compared to previous modeling studies, our simulations are well within modeling estimates and in the lower range of AIS volume responses (Figure 5a). No simulation reaches the lowest limit of 11 mSLE set by DeConto et al. (2021, orange line), and just a few reach the lowest limit of 7 mSLE set by DeConto et al. (2021, blue line). Results from Yan et al. (2016, pink line) and Golledge et al. (2017, green line) are closer to our upper limit, whereas those of Berends et al. (2019, purple line) and Dolan et al. (2018, red line) are inside the range of our simulations and de Boer et al. (2015, brown line) simulates a lower contribution. In comparison with GIA studies of Hollyday et al. (2023, grey line) and Richards et al. (2022, yellow line), our results show agreement within the lowest bounds.

Figure 6 shows the ice-collapse probability for every AOGCM forcing applied (red: high probability of collapse, blue: low probability), determined at each point as the fraction of simulations showing a collapse at that point from the ice-sheet model ensemble. All cases (with the exception of MIROC4m) show a collapsed WAIS, though in some cases this retreat is more pronounced (COSMOS) than in other cases (NorESM1-F). In the Wilkes basin, three AOGCM climates induce a retreat of the marine regions, though with different probabilities: low to medium in CESM1.0.5 and high in NorESM-L and EC-Earth3.3. The simulated sea-level contribution of these cases are: $6.0^{+1.8}_{-1.3}$ (CESM1.0.5); $8.9^{+0.2}_{-0.3}$ (EC-Earth3.3); $6.8^{+0.1}_{-0.1}$ (NorESM-L). Totten glacier shows a slight retreat only for CESM1.0.5. Some regions of the EAIS close to the Filchner-Ronne ice shelf also retreat in some cases, especially for EC-Earth3.3 and CCSM4-UofT. Generally less extended ice sheets lead to lower volumes, though this is not always the case (see simulations with HadGEM3 and MRI-CGCM2.3 forcing in Figure 5).

In order to assess the spatial origin of the mass loss for every AOGCM forcing we plot the mean ice thickness anomaly between the simulated PlioMIP2 and PD state (Figure 7). The ice thickness anomaly is nearly always negative (red colors) in the WAIS, since it has collapsed in that region. Even MIROC4m shows a negative thickness anomaly, though smaller in magnitude. In contrast, the EAIS presents more complex behavior depending on the AOGCM forcing. A warmer atmosphere enhances precipitation. Thus, the interior of the EAIS gains volume for some AOGCMs (CCSM4-UofT, HadGEM3, IPSLCM5A, IPSLCM5A2). Nonetheless, sufficiently high ocean temperatures can induce a grounding-line retreat in the Wilkes basin. This is the case for simulations forced by CESM1.0.5, EC-Earth3.3, and NorESM-L. Simulations with COSMOS, NorESM1-F and MRI-CGCM2.3 show a slightly negative anomaly in the coastal regions of EAIS. Although it does not propagate further inland,

it seems to compensate for inland accumulation, leading to a value close to zero. This spread in the EAIS and more specifically in the Wilkes basin points to an important role of the applied boundary conditions in the model response.

## 3.2 Tipping point analysis

### Climatic forcing

We find in our study three potential sites prone to collapse: The WAIS through the Amundsen region, the Wilkes basin, and, on a smaller scale, the Totten basin. We find that oceanic temperatures are the main forcing defining the ice extent of marine basins (Figure 8). In the case of the Amundsen region (Fig. 8a), we observe that all simulations show a collapsed WAIS with one exception, the MIROC4m model. Though this result is not realist in terms of sea-level equivalent, pointing at an AIS bigger than today, it shows interesting results in terms of tipping points in the Amundsen sea.

It is clear from Fig. 8a that even small temperature increases can lead to a collapse of the WAIS, but that changes in precipitation can play a key role for low temperatures. Since we want to focus on the tipping point and thus the minimal oceanic temperature anomaly that leads to a collapse of the Amundsen Sea embayment, we focus on the four models that do not exceed 1 K of oceanic anomaly: COSMOS (0.44 K), IPSLCM5A (0.92 K), IPSLCM5A2 (0.86 K) and MIROC4m (0.58 K). Note that this anomaly has been computed for each basin (Fig. S1) at bedrock depth. By plotting the relative precipitation against the thermal forcing anomaly (Fig. 8d), we find that MIROC4m shows a relative precipitation anomaly close to PD values, whereas the IPSLCM5A and IPSLCM5A2 precipitation anomaly lies around 85% of PD precipitation. Especially notable is the case of COSMOS, where the temperature anomaly is lower than MIROC4m, but also the precipitation, around 78%. Thus, we see that a thermal forcing below 0.5 K can lead to a collapse of the WAIS if precipitation stays below 80% of PD. Above 1 K anomaly we always find a collapsed WAIS even for precipitation rates close to PD (EC-Earth3.3). Nonetheless, it is important to mention that around 20-40% (Fig. 6) of the MIROC4m simulations show a collapsed Amundsen embayment.

We redo the same analysis with the Wilkes basin to investigate the tipping points that can lead to a collapse (Fig. 8b). Since the Wilkes basin also lies on a retrograde bedrock, we assume that the oceanic thermal forcing is the main trigger. We find that the three AOGCMs that cause a collapse, namely EC-Earth3.3 (80-100 %), CESM1.0.5 (40-60 %) and NorESM-L (80-100 %) simulate an oceanic anomaly above 3 K. Surprisingly, the CESM1.0.5 model, which has the highest thermal forcing anomaly, yields the highest uncertainty in the retreat (around 50%, Fig. 6). This can be explained partially by the precipitation anomaly, that is three times larger than PD rates (Fig. 8e). EC-Earth3.3 and NorESM-L have similar thermal forcing and precipitation anomalies (around 130% of PD rates) and thus lead to similar results. Therefore, we conclude that a warming above 3 K can lead to an irreversible retreat of the Wilkes basin. Nonetheless, this retreat can be somewhat mitigated by basin-wide enhanced precipitation rates as seen in CESM1.0.5. This suggests an important role of ice dynamics. Hence, in the next section we will analyze the potential role of ice dynamics for CESM1.0.5.

Finally we focus on Totten glacier, since it also shows signs of potential instabilities for CESM1.0.5 (Fig. 6). Redoing the same analysis for that basin (Fig. 8c,f), we find that CESM1.0.5 simulates the lowest ice extent in Totten due to a thermal forcing anomaly above 8 K. The other models do not show a significant retreat (extension above 90% of PD), even for thermal

forcings close to 4 K. Thus, we conclude that for the Totten glacier, oceanic anomalies well above 4 K are needed to induce a retreat of the grounding line there.

**Ice dynamics**

Since we show that some basins collapse with certain probability for forcing from some AOGCMs, we focus our attention on the role of the ice dynamics in the ice retreat (Fig. S7). We plot the three main parameters influencing the ice flow that we permuted in our simulations (Enhancement factor $E_f$, friction law exponent $q$ and friction coefficient $c_f$) for the two AOGCMs that showed a certain probability of collapse (CESM1.0.5 in the Wilkes and Totten basin, and MIROC4m in the Amundsen Sea basin). For CESM1.0.5, we could not find any relationship between a Wilkes collapse and the dynamic configuration, except that lower enhancement factors simulated a more pronounced retreat than higher enhancement factors. On the other hand, for the Totten glacier we find that simulations with higher enhancement factors ($E_f$=5) never collapse, whereas simulations with lower values ($E_f$=3) always collapse. Intermediate values ($E_f$=4) show a regime with both states. Finally, no clear relationship is found for the MIROC4m model in the Amundsen Sea region, except that neither an enhancement factor of $E_f$=3 nor a linear friction law ($q = 1$) collapse.

### 3.3 Bedrock experiments

Some additional simulations were performed to test the effect of different topographic initial conditions on the final results. To avoid running the complete ensemble again, we took the ensemble parameters ($c_f$, $E_f$ and q) which simulated the closest value to the mean for every AOGCM. We performed an additional set of simulations by imposing the Pliocene topography and ice-thickness configuration from PRISM4 (Dowsett et al., 2016). PRISM4 surface elevation is illustrated in Fig. S6. Figure 9 shows the surface elevation of the simulated AIS. In this case, all the simulations show a collapsed WAIS as well as Wilkes and Totten basin. These results are more in agreement with the reconstructions used for the PRISM4 boundary conditions and the highest range of sea-level estimates (sea-level contributions from 15 to 25 meters). Nonetheless, as we will discuss further, these results are biased towards a collapsed state, since regrowth on retrograde bedrock slopes is hampered. The existence of positive feedback mechanisms on marine retrograde bed slopes creates hysteresis behavior.

## 4 Discussion

### 4.1 Comparison with previous studies

We have presented a large ensemble forced with different mPWP climatologies. DeConto and Pollard (2016) and DeConto et al. (2021) also performed a large ensemble analysis but only explored the relationships between ocean temperature and sub-ice-shelf melt rates, hydrofracturing and maximum rates of marine-terminating ice-cliff failure. Yan et al. (2016) used an ensemble to investigate parameters that affect the climatic conditions, rather than ice dynamics. de Boer et al. (2014) and Dolan et al. (2018) include several climatic outputs and ice-sheet models. Nonetheless, only one dynamic configuration was chosen

for every ice-sheet model. Here we aimed to consistently investigate the role of uncertainties in both the mPWP climatology by testing different AOGCMs as well as ice dynamics.

In total we simulate an Antarctic sea-level contribution of less than 10 meters if we start from PD conditions and use the PD topography (Figure 5). Our results are in general agreement with many studies that start with PD initial conditions or evolve transiently towards the mPWP (de Boer et al., 2014; Yan et al., 2016; Golledge et al., 2017; Dolan et al., 2018; Berends et al., 2019). Our simulations differ greatly with the studies from DeConto and Pollard (2016) and DeConto et al. (2021) due to their inclusion of the MICI mechanism. Without MICI, those studies only show a collapse of the WAIS corresponding to 3 mSLE (Fig. S6a,b). However, it is worth mentioning, that those studies applied an oceanic anomaly of only 2 K warming with respect to PD at the mPWP. As shown in our study, with such a forcing we would not simulate a collapse of the Wilkes or Totten glacier retreat either, since at least a 3 K oceanic warming anomaly is needed. Furthermore, Crawford et al. (2021) showed that the applied retreating rate for small cliffs was overestimated in DeConto and Pollard (2016). Other studies that achieve a collapse of the Wilkes basin do it either by increasing oceanic temperatures (Yan et al., 2016; Golledge et al., 2017) or by adding melt at the grounding line (Golledge et al., 2017). Though not focused on the Pliocene, the ABUMIP experiments showed that the removal of ice shelves also leads to substantial ice loss in the Wilkes basin for most models, showing that it is a highly vulnerable and uncertain region (Sun et al., 2020). Our results support a collapse of the Wilkes basin for an oceanic anomaly of 3 K and a retreat of the Totten glacier for an oceanic anomaly above 4 K. Nonetheless, high precipitation rates can hamper this retreat.

Since our Antarctic sea-level contributions do not exceed 10 mSLE, our simulations do not support a global sea-level contribution of more than 20 mSLE as suggested by some reconstructions (Dumitru et al., 2019; Hearty et al., 2020). Nonetheless, recent work done with geodynamic models suggest a lower contribution at the mPWP than proxy data. These models simulate dynamic topographic changes on specific domains, namely the Patagonian region (Hollyday et al., 2023), the Australian region (Richards et al., 2022) and the Atlantic shoreline (Moucha and Ruetenik, 2017). The main advantage compared to proxy data is that processes that are difficult to assess on in-situ measurements and have a big impact, such as geostatic uplift, can be considered. These results are then compared to proxy measurements from that region to assess the reality of their simulation. The new sea-level estimates reduce the global sea-level contribution significantly: $17.5 \pm 6.4$ mSLE (Hollyday et al., 2023); $16.0 \pm 5.5$ mSLE (Richards et al., 2022); 15 mSLE (Moucha and Ruetenik, 2017). Assuming that Greenland was almost fully melted ($\sim 7.4$ mSLE, Morlighem et al. (2017)), with such a revised sea-level reconstruction, our results are inside the geological constraints if Wilkes basin collapsed via high oceanic thermal forcing or with low precipitation rates, as in MRI-CGCM2.3 (Table 1 in SM). Richards et al. (2022) even go one step further and argue that the impact of the proposed MICI mechanism (DeConto and Pollard, 2016; DeConto et al., 2021) is overestimated. Though this is not the scope of our work, these new results could highlight the need for new mPWP boundary conditions for AOGCMs, mainly a larger and thicker AIS than previously thought.

Although not focused on the mPWP, the study of Garbe et al. (2020) shows a threshold of the Wilkes basin between 4 to 6 K of warming relative to pre-industrial levels for the atmosphere (equivalent to 1.5-2.5 K in the ocean in their study). The

Totten basin retreats in their experiment with an atmospheric anomaly of 7 K (close to 3 K of oceanic warming). Nonetheless, as pointed out by their study, this threshold is highly sensitive to structural model dependence.

In our study we do not find a clear distinction between our ensemble ice-sheet dynamics-related parameters and the simulated ice extent or ice volume. Simulations forced with CESM1.0.5, simulate a slightly more retreated Totten basin for low enhancement factors (Fig. S7). We believe this is a consequence of the simulated PD state rather than the modelled ice dynamics, since we did not use any constraint metrics in the EAIS. In the MIROC4m model we find that a WAIS collapse is more likely to occur for high enhancement factors and low friction exponents, which promotes faster ice flow. In summary, although we observe some trends associated with the dynamic configuration for CESM1.0.5 and MIROC4m, no clear relationship can be found between ice extent and the ensemble dynamical parameters. Our analysis allows us to assess the sea-level uncertainties that arise from dynamical configuration and climatologies. We find that the climatologies yield a larger uncertainty ($\sim$7 mSLE) than that resulting from the dynamic configuration if parameters are constrained with PD observations. Dolan et al. (2018) obtain more than 10 mSLE between different ice-sheet models, whereas we obtain less than 2 mSLE differences for simulations which are not close to tipping, and up to 5 mSLE differences for CESM1.0.5 due to the proximity of Wilkes basin to tipping or not (error bars Fig. 5).

## 4.2   Forcing limitations

In our study, the transient character of the climate system was neglected for the sake of simplicity and as well as the poor knowledge on the transient forcing. Instead, we forced our model towards a steady mPWP state for an ensemble large enough to be statistically significant (more than 30 simulations) for 12 different mPWP conditions. This approach permits us to assess the Antarctic tipping points starting from PD conditions as well as the impact of the uncertainty associated with state-of-the-art equilibrium mPWMP climatic conditions. This experimental setup goes in line with other studies, allowing for a similar comparison (Yan et al., 2016; DeConto and Pollard, 2016; DeConto et al., 2021). However, assuming a constant warming may lead to overestimation of sea-level contributions since we impose a warm climate over longer timescales than for a transient experiment. For instance, as shown by Stap et al. (2022), the simulated Antarctic sea-level contribution at the Miocene is lower for a transient forcing than for a constant forcing leading to steady state. To our knowledge, only one study has simulated the transient evolution of the AIS under the Pliocene (Berends et al., 2019). The transient climate forcing they used did not reach the necessary conditions to lead to a retreat in the Wilkes basin, and thus produced a relatively low sea-level contribution (Fig. S6c)

It is important to mention that exceeding a tipping point does not mean that the ice sheet will collapse immediately, but rather that it has reached the threshold temperature by which a retreat will be induced and further amplified by MISI. By plotting the one dimensional evolution of the WAIS (Fig. S5), we observe that the WAIS collapse usually occurs with a lag of 1000-5000 years from the application of the forcing. In some cases it can reach up to 25000 years. MISI is not only a matter of the oceanic temperature threshold, but also depends on the grounding-line position and the thermal forcing at this location, as well as precipitation. Thus, a transient character in the forcing could avoid certain ice collapses if the warming is not sufficiently

long. Other factors, such as ice dynamics, could also delay (or accelerate) the grounding-line position reaching a pronounced retrograde bedrock that leads to a full collapse of the WAIS or other marine basins.

Another limitation in our study is the initial topographic boundary condition. In order to overcome this problem we performed additional experiments starting from the topography and ice-sheet thickness reconstructed from PRISM4 conditions (Fig. 9). Our sea-level estimates then shift towards the high-range estimates, between 15-25 mSLE. Such an experiment was performed in the study of Dolan et al. (2018) and de Boer et al. (2014). Their results also show that starting from PRISM4 conditions leads to higher sea-level contributions and a less extended AIS during the mPWP. This result is expected since a smaller ice sheet has warmer temperatures due to the melt-elevation feedback, captured in our experiments through an atmospheric lapse-rate factor. In addition, growing back on a retrograde marine basin needs a strong decrease in ocean temperature due to the hysteresis behavior of the ice sheet. Runs that are initialized with PRISM4 conditions show an Antarctic sea-level estimate up to 20 mSLE.

The mPWP was preceded by a large global glaciation during Marine Isotope Stage M2, ca. 3.3 MaBP (Rohling et al., 2014; Stap et al., 2016). During that period, the AIS evolved towards a modern-like configuration (Berends et al., 2019). Therefore, starting from PD initial conditions can help to assess the realism of the simulated mPWP from the AOGCMs. Our model only simulates a retreat in the Wilkes basin, supported by reconstructions, for three out of twelve AOGCM models.

Our forcing strategy based on an anomaly-snapshot method (i.e. one constant climatic snapshot from each AOGCM) ignores certain climate interactions that could be relevant to the system. We take into account the surface melt-elevation feedback by employing the aforementioned lapse-rate factor and albedo-melt feedback within our ITM parameterisation. However, these interactions could be improved with a spatially varying lapse-rate factor computed from AOGCM temperature and elevation data (Crow et al., 2024). Nonetheless, probably one of the most important feedbacks not considered here is the effect of freshwater flux release from the AIS into the Southern Ocean. Results from Sadai et al. (2020) show that accounting for Antarctic ice discharges increases subsurface Southern Ocean temperatures. However, Bintanja et al. (2015) showed that ice-shelf melt leads to a cooling of the Southern Ocean and an expansion of sea ice area. This points to the need for a more profound understanding of ice-ocean related processes within models.

A more sophisticated approach would include direct coupling between an AOGCM and our ice-sheet model. However, besides more computational resources, this would require constraints not only on our ice-sheet model parameters, but also on those of the AOGCM. The work of Berends et al. (2019) is a good example of a coupled ice-sheet model based on a matrix method. However, in order to run these simulations at global scales, one trade off is a lower ice-sheet resolution (40 km). This is a potential explanation as to why they do not simulate a retreat in the East Antarctic region. Here we aim to obtain a more profound understanding of processes related to ice dynamics in part through a higher spatial resolution (16 km).

Finally, there exist additional sources of forcing uncertainties which have not been taken into account, such as geothermal heat flow or the Earth rheology. On one hand, assessing the geothermal heat flow at the PD represents a source of uncertainty (Burton-Johnson et al., 2020), thus its value during the mPWP represents a major unknown. Earth rheology in this study was considered homogeneous for the whole AIS based on the the elastic lithosphere-relaxed asthenosphere method (Le Meur and

Huybrechts, 1996). Our study did not focus on the role of GIA on our simulations, however new model implementations are planned in future work with Yelmo with a new GIA model which includes lateral variability (Swierczek-Jereczek et al., 2023).

## 4.3 Model limitations

As shown by Pattyn et al. (2013), high resolution is needed at the grounding line to simulate accurate grounding-line migrations. Ice-sheet models use different techniques at the grounding line to compensate for coarse resolution, such as flux conditions (Schoof, 2007; Tsai et al., 2015) or scaling friction at the grounding line by the grounded ice fraction. In our study we use the latter technique, which has been shown to simulate realistic grounding-line migrations on idealized domains (a thorough description is presented by Robinson et al. (2020)). We also ensure that effective pressure, which enters the basal friction equation, tends to zero as the ice thickness approaches flotation (Leguy et al., 2014). Nonetheless, grounding-line representation remains a source of uncertainty that can strongly influence the retreat of marine based glaciers prone to MISI.

Another source of uncertainty is the melting at the grounding line. Observations have established that the ocean-induced basal melting is highest close to the grounding line and decreases towards the ice-shelf front (Adusumilli et al., 2020). Ice-sheet models use different approaches which typically range from no ocean-induced melting to partially ocean-induced melting (Seroussi and Morlighem, 2018; Leguy et al., 2021). In many coarse resolution ice-sheet models (more than 2 km resolution at the grounding line), no melting is applied directly at the grounding line since it can lead to overestimation of sub-shelf melting (Seroussi and Morlighem, 2018). Nonetheless, recent studies suggest that at high spatial resolution, applying melting at the grounding line via a flotation criterion may be more accurate since it is less resolution dependent (Leguy et al., 2021; Berends et al., 2023). This could suggest that our results correspond to a lower limit since no melting is applied at the grounding line in our experiments. We expect that by adding melting at the grounding line, the collapse of the Wilkes basin would have been more likely for those AOGCM climates with lower oceanic thermal forcing. Basal melting representation remains a fundamental source of uncertainty which needs further investigation.

A final source of uncertainty comes from the unknown basal conditions. Here we used a spatially constant friction coefficient scaled with the bedrock depth to favour more sliding at deeper bedrocks. Another common approach is to compute friction coefficients through an optimization procedure aiming at minimising the errors in ice thickness with respect to observations (Lipscomb et al., 2019, 2021). We performed an additional set of experiments following this approach by optimizing over 30000 years towards PD conditions. Our Pliocene simulations in this case showed similar results to those achieved for depth-dependent friction coefficients except for the MIROC4m case, where the ice extent is lower for optimized friction coefficients (Fig. S8). However, this is not surprising at all, since the optimized simulations have not reached equilibrium. If we let the optimized experiments run for additional 30000 years with PD forcing we see that the ice volume decreases for seven of nine cases indicating a WAIS collapse (Fig. S9). Such a trend in ice volume for optimized friction coefficients has been observed in other ice-sheet models even for shorter timescales (Seroussi et al., 2020; Coulon et al., 2023). Thus, we believe that the basal friction optimization approach is not valid for long timescales and it is more appropriate to maintain our methodology. Our simulations with a homogeneous friction coefficient produce satisfactory results in terms of RMSE of ice thickness and surface velocities, that are comparable to those of other groups in the context of ISMIP6 (Fig. S3). Furthermore, there is no

a priori reason to believe that optimized friction coefficients for PD would have been the same for the mPWP. Our approach has the benefit that basal friction adapts to changes in ice thickness and effective pressure as a result of changes in the mPWP boundary conditions with respect to present day. Therefore, we believe that for our study, it is more beneficial to use a simple parameterization as in other paleo-studies (Quiquet et al., 2018), rather than optimized friction coefficients. A potential future improvement could be to include an active sediment mask to account for changes in erosion, which can change the bed roughness.

## 5  Conclusions

Here we investigated the AIS response to mPWP conditions to assess its sea-level contribution during the mPWP and the potential tipping points that our ice-sheet model exhibits under mPWP scenarios. A way to gain insight into tipping-point behaviors of ice sheets would be to perform an intercomparison between different ice-sheet models and analyze different sources of uncertainty, such as grounding-line basal melt, basal friction at the grounding line or resolution, among others. In this study we aimed to contribute to this discussion by testing dynamic sources of uncertainty in the Yelmo ice-sheet model under different mPWP climatic forcings in the framework of the PlioMIP2 project. We have identified that the WAIS exhibits a tipping point for an oceanic warming of 0.5 K, as long as regional precipitation remains below that of PD. When the oceanic warming reaches 1 K anomaly, even precipitation similar to today's or higher is unable to prevent a MISI. In the Wilkes basin, a retreat occurs when the oceanic warming reaches 3 K. However, we have observed that high precipitation, up to three times higher than today, can potentially prevent such a retreat. Additionally, we have found that the Totten glacier can also retreat, but only under high oceanic warming conditions at least above 4 K oceanic anomaly. In addition, we explored the initialization of the model with an ice-sheet thickness derived from PRISM4. This initialization resulted in a lower AIS in terms of both ice volume and extent due to starting from already retreated marine basins. Consequently, the model initialized with the PRISM4 ice-sheet thickness displayed persistent differences in simulated AIS characteristics compared to other initializations.

Finally, our mean simulated sea-level contributions for every AOGCM ranged from $2.7_{-0.4}^{+0.1}$ mSLE to $8.9_{-0.3}^{+0.2}$ mSLE considering the whole ensemble starting from PD conditions, and 15.5 mSLE to 25.6 mSLE when starting from PRISM4 conditions. If only the WAIS collapses, sea-level contributions ranges from $2.7_{-0.4}^{+0.1}$ mSLE to $7.0_{-0.1}^{+0.1}$ mSLE. If only the Wilkes basin collapses, sea-level contributions range from $6.0_{-1.3}^{+1.8}$ mSLE to $8.9_{-0.3}^{+0.2}$ mSLE. The contributions stating from PD conditions are in agreement with geological constraints which do not exceed global sea-level stands above 20 mSLE. However, the collapse of the Wilkes basin is a necessary condition in order to achieve Antarctic sea-level rises above 7 mSLE. Ultimately, the MICI mechanism is not a necessary condition for a collapse of the Wilkes basin, since high oceanic temperatures can also lead to such a collapse. Our results reinforce the hypothesis that crossing several Antarctic tipping points is necessary for large sea-level high stands to be obtained at the mPWP.

## Code and data availability

Yelmo is maintained as a git repository hosted at https://github.com/palma-ice/yelmo under the licence GPL-3.0. Model documentation can be found at https://palma-ice.github.io/yelmo-docs/. The results used in this paper will be made available on Zenodo once published.

*Author contributions.* JB carried out the simulations, analyzed the results and wrote the paper. All other authors contributed to designing the simulations, analyzing the results and writing the paper.

*Competing interests.* At least one of the (co-)authors is a member of the editorial board of *Climate of the Past*.

*Acknowledgements.* This project is TiPES contribution #246: This project has received funding from the European Union's Horizon 2020 research and innovation programme under grant agreement No 820970. A.R. received funding from the European Union (ERC, FORCLIMA, 101044247). This research has also been supported by the Spanish Ministry of Science and Innovation project MARINE (grant agreement No PID2020-117768RB-I00). Simulations were performed in Brigit, the HPC of the International Campus of Excellence of Moncloa, funded by MECD and MICINN.

| Parameter | Units | Values | Description |
|---|---|---|---|
| $E_f$ | – | 1-5 | Enhancement factor |
| q | – | 0.0,0.2,1.0 | Friction law exponent |
| $c_f$ | – | 0.1-1.0 | Basal friction coefficient |
| $u_0$ | m yr$^{-1}$ | 100 | Basal velocity regularization term |
| $\kappa_t$ | m yr$^{-1}$ Pa$^{-1}$ | 0.0025 | Calving scaling parameter |
| $\omega_2$ | – | 25 | Calving eigenvalue weighting coefficient |
| $\rho_w$ | kg m$^{-3}$ | 1000 | Pure water density |
| $\rho_{sw}$ | kg m$^{-3}$ | 1028 | Sea water density |
| $\rho_i$ | kg m$^{-3}$ | 917 | Pure ice density |
| $L_i$ | J kg$^{-1}$ | 3.34 10$^5$ | Latent heat of fusion ice |
| $c$ | W m$^{-2}$ | -55 | Short-wave radiation and sensible heat flux constant |
| $\lambda_{srf}$ | W m$^{-2}$ K$^{-1}$ | 10 | Long-wave radiation coefficient |
| $\gamma_{quad-nl}$ | m yr$^{-1}$ | 14500 | Oceanic heat exchange velocity |
| $c_{po}$ | J Kg$^{-1}$ K$^{-1}$ | 3974 | Specific heat capacity of ocean mixed layer |
| $\lambda_1$ | °C PSU$^{-1}$ | -0.0575 | Liquidus slope |
| $\lambda_2$ | °C | 0.0832 | Liquidus intercept |
| $\lambda_3$ | °C m$^{-1}$ | 7.59 10$^{-4}$ | Liquidus pressure coefficient |

**Table 1.** Table summarizing the model parameters.

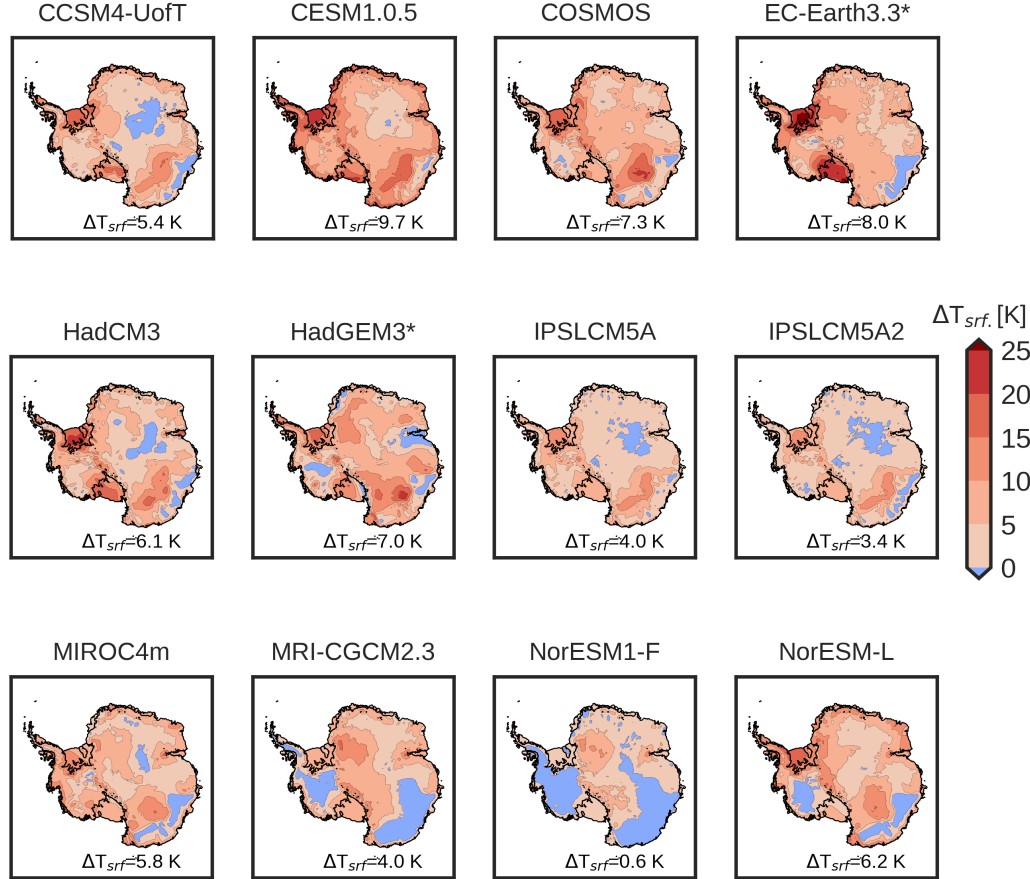

**Figure 1.** Sea-level temperature anomaly fields of the employed PlioMIP2 AOGCMs. Negative values (blue colors) represent a colder than PD surface temperature. Positive values (red colors) indicate a warmer than PD surface temperature. Numbers on the lower right corner shows the mean temperature anomaly inside the PD Antarctic domain (contour lines of the Antarctic grounding-line and ice shelves). CMIP6 models are marked with an asterisk.

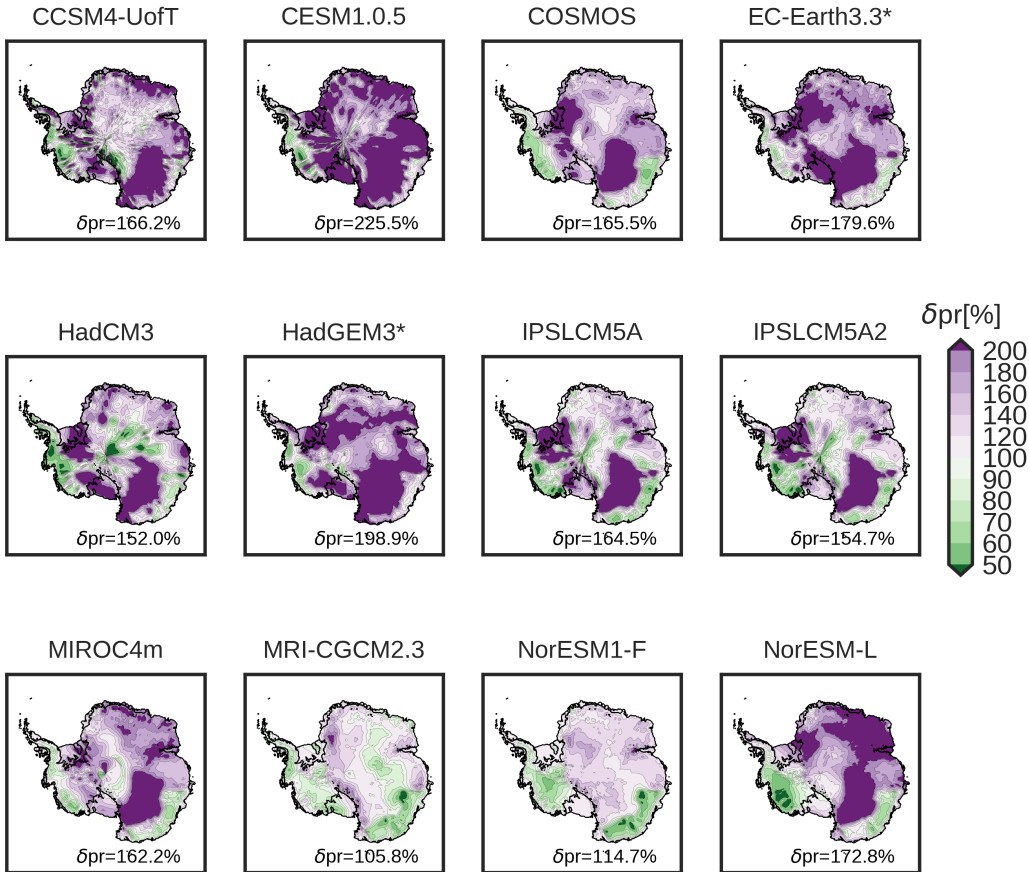

**Figure 2.** Relative precipitation anomaly fields of the employed PlioMIP2 AOGCMs at sea-level elevation. Values below 100% (green colors) represent a drier climate (less precipitation than PD). Values above 100% (purple colors) indicate more precipitation than in the PD. Numbers on the lower right corner shows the mean relative precipitation anomaly inside the PD Antarctic domain (contour lines of the Antarctic grounding-line and ice shelves).

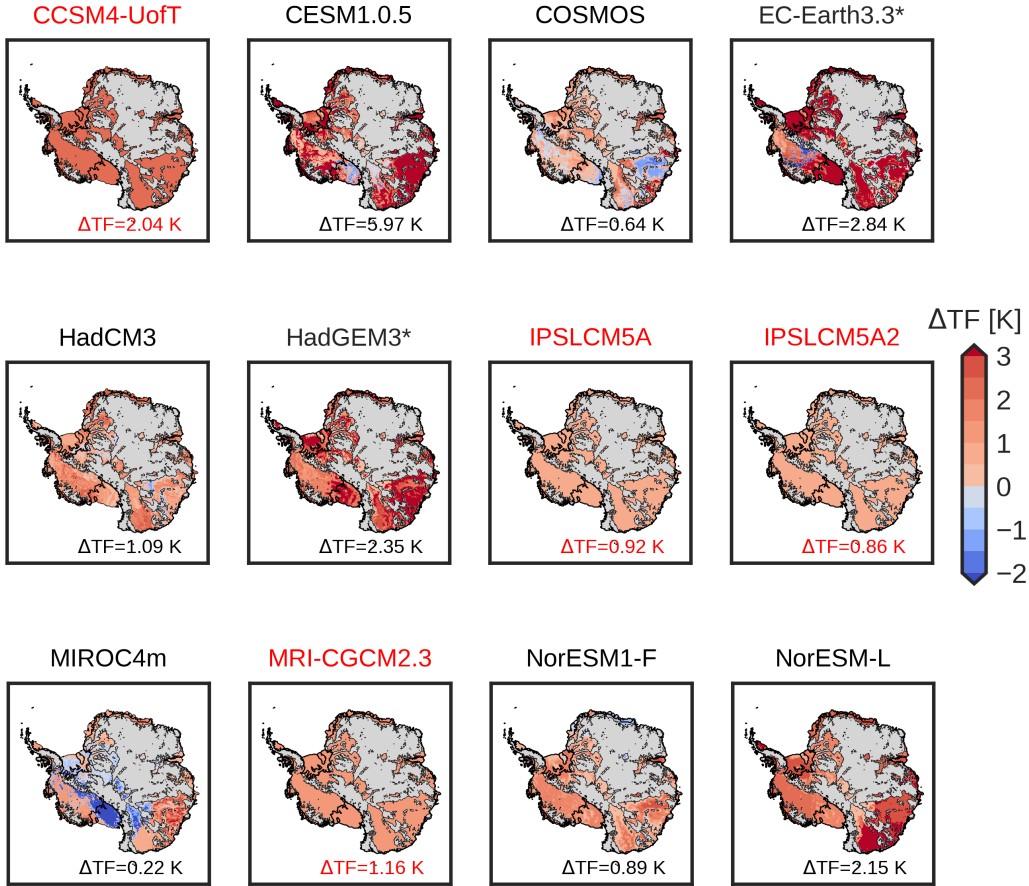

**Figure 3.** Ocean thermal forcing temperature anomaly fields at the ice-ocean interface of the employed PlioMIP2 AOGCMs. Positive values (red colors) indicate a warmer bed-ocean temperature than in the PD. Gray colors indicate bedrock above sea level (based on PD topography) and thus, with no ice-ocean interaction. The number on the lower right corner shows the mean bed-temperature anomaly inside the PD Antarctic domain (contour lines of the Antarctic grounding-line and ice shelves). Models in red are AOGCMs that did not provide any ocean field. The inferred ocean field was obtained as a mean of the atmospheric temperatures scaled by a fraction of 1/4 (Taylor et al., 2012).

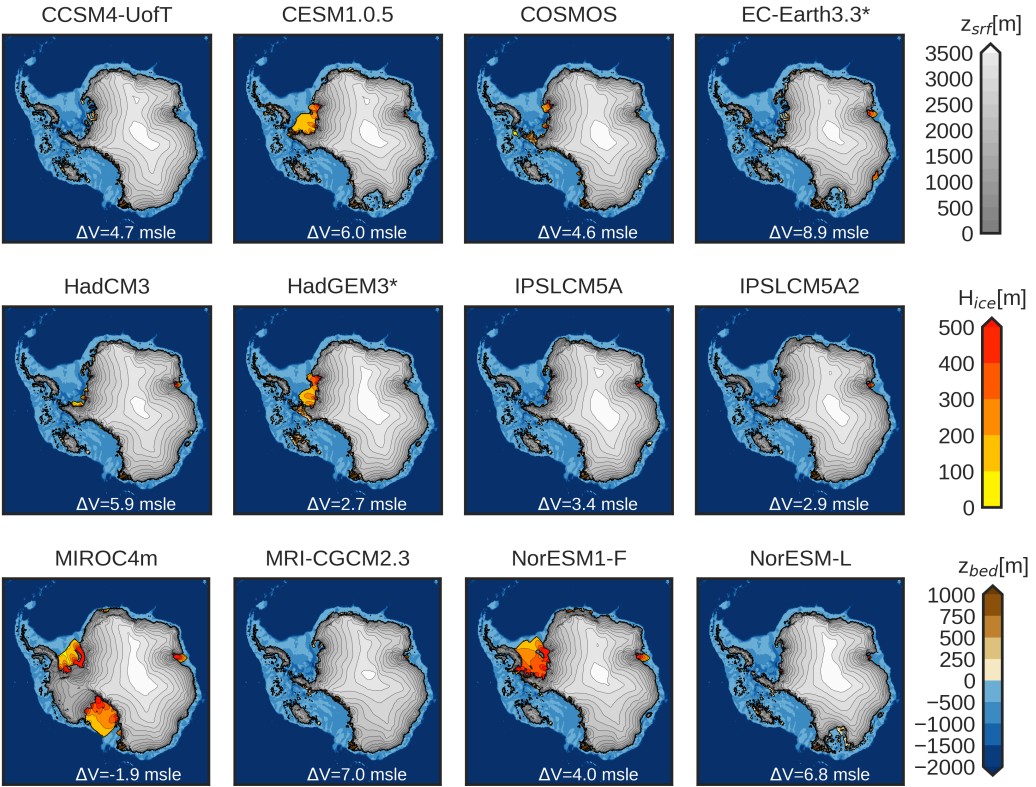

**Figure 4.** Surface elevation (gray), floating ice thickness (orange) and bedrock elevation (brown/blue) of the simulation closest to the mean ice volume and ice extent of the ensemble for every AOGCM starting from PD bedrock conditions. White number in the bottom corner represents the sea-level rise with respect to the PD state.

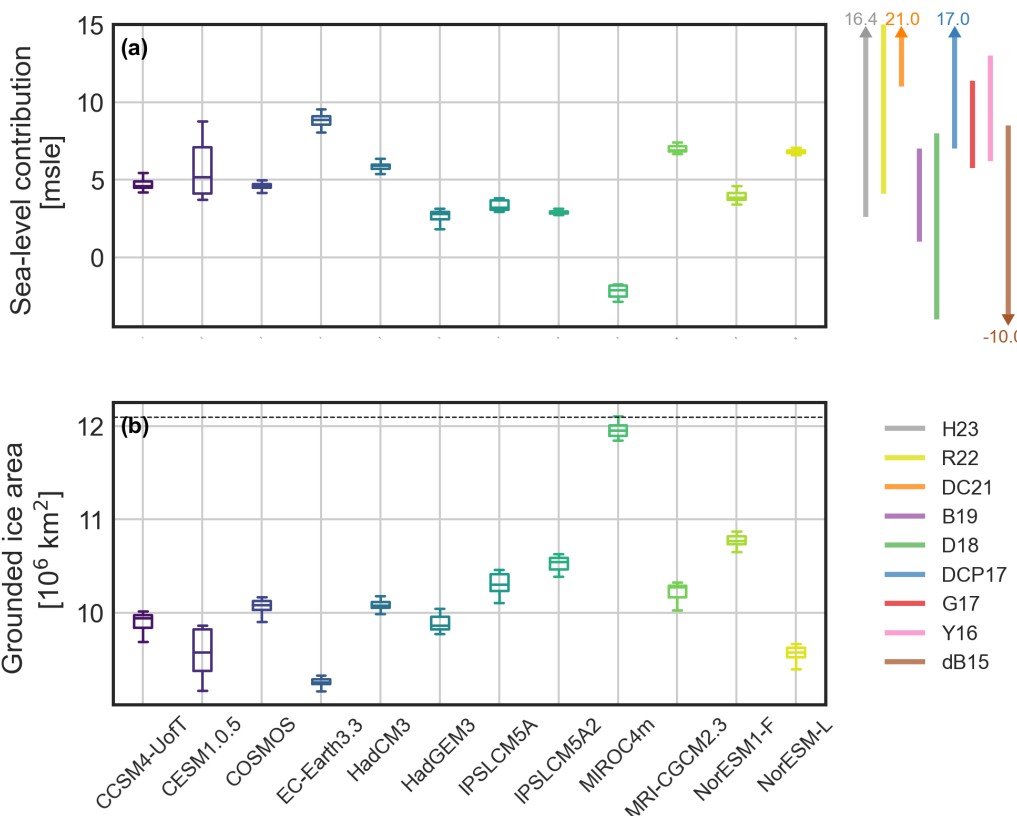

**Figure 5.** Boxplot of the simulated (a) sea-level contribution (positive/negative numbers indicate a lower/higher ice volume); (b) grounded ice extent for every AOGCM. The scatter-point shows the mean values of the ensemble. The error bars represent the lowest/highest simulated AIS state starting from PD conditions. Light shaded colors at the right show the sea-level uncertainty ranges from the studies of deBoer et al., (2015, brown); Yan et al., (2016, pink); Golledge et al., (2017, red); DeConto and Pollard (2017, blue); Dolan et al., (2018, green); Berends et al., (2019; purple); DeConto et al., (2021, orange); Richards et al., (2022; yellow); Hollyday et al., (2023; grey). The dashed black line in (b) represents the PD grounded ice extent.

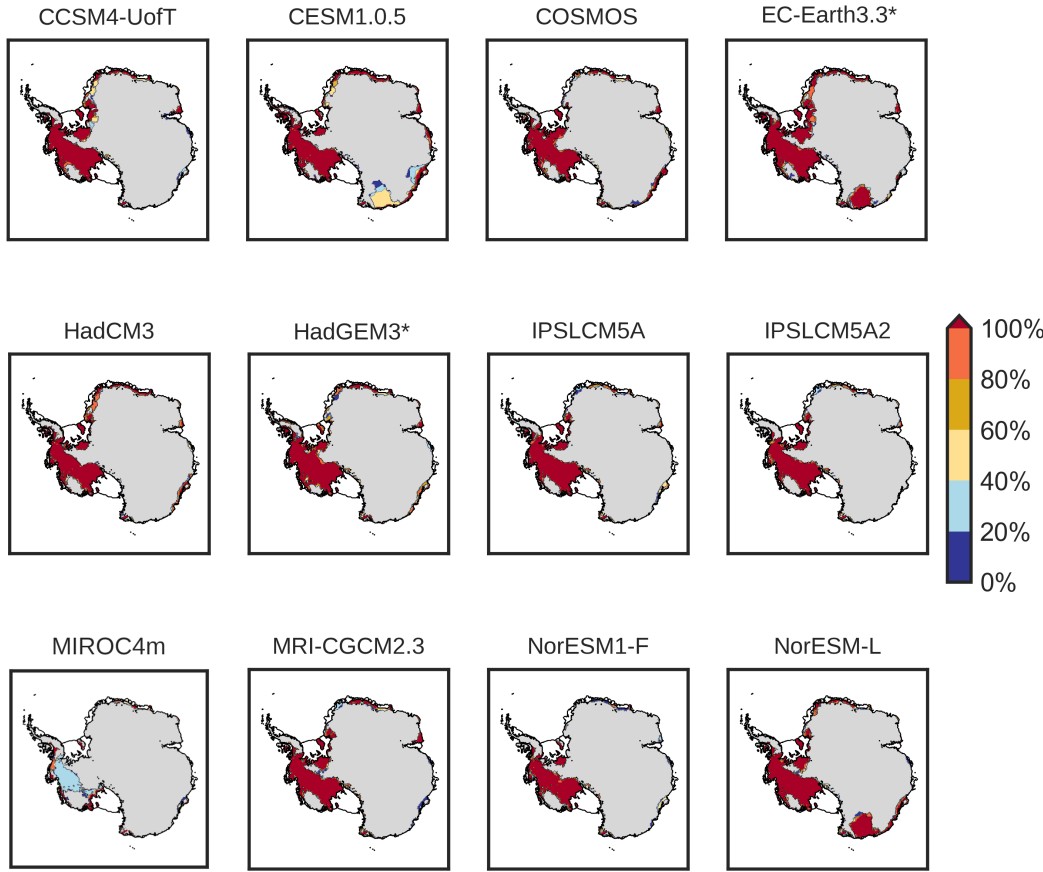

**Figure 6.** Probability of ice collapse of the ensemble for every AOGCM. Red colors indicate a 100% probability of collapsed regions. Gray colors show grounded ice for all the ensemble simulations.

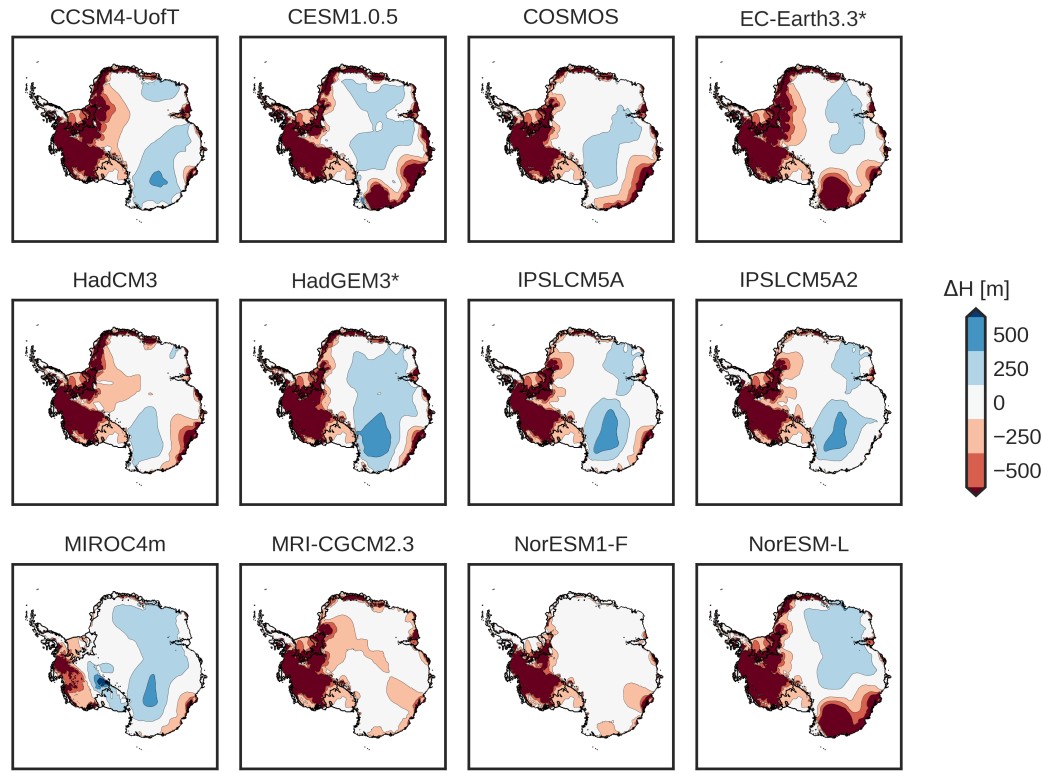

**Figure 7.** Mean ice-thickness anomaly between the mPWP state and the PD state. Positive/negative numbers (blue/red) represent a thicker/thinner ice column than the simulated PD.

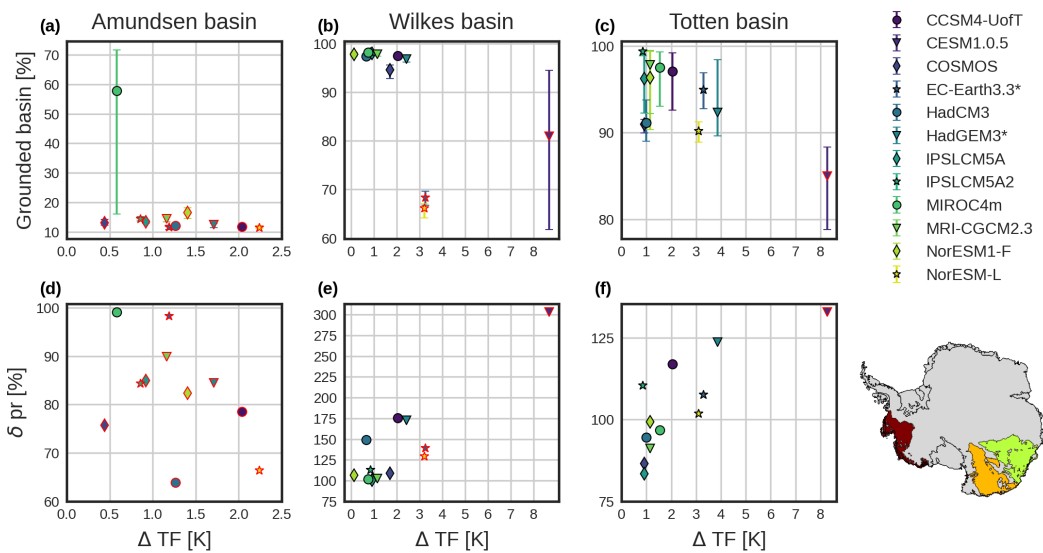

**Figure 8.** Scatter plot of grounded simulated AIS ice area at the mPWP (in percentage of the marine basin as in Fig. S1) with respect to the thermal forcing anomaly for (a) Amundsen basin; (b) Wilkes basin; (c) Totten basin in the retreated regions (Basins in Fig. S1). The error bars represent the lowest/highest simulated AIS state. (d- f) Same as a-c but for the relative precipitation anomaly relative to PD. Red borders represent either collapsed marine basins or more retreated than the rest of AOGCMs. In the bottom right corner the regions of interest are highlighted: Dark-Red: Amundsen. Orange: Wilkes. Green: Totten.

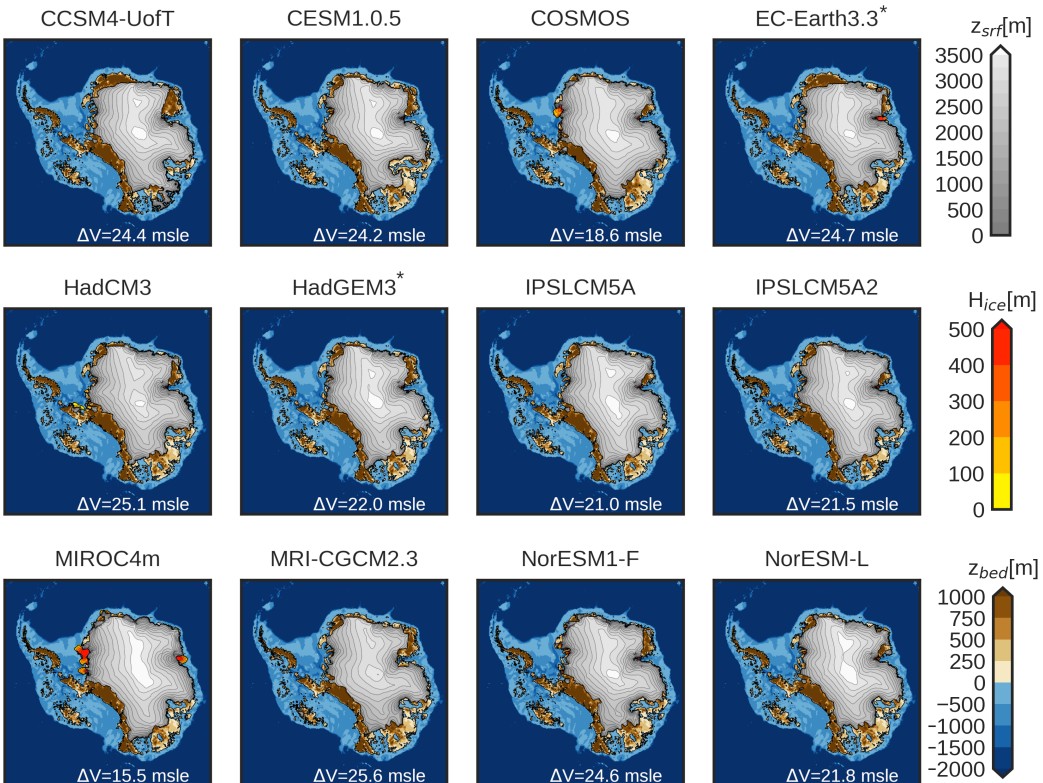

**Figure 9.** Surface elevation (gray), floating ice thickness (orange) and bedrock elevation (brown/blue) of the simulation closest to the mean volume and extension of the ensemble for every AOGCM forcing, starting from PRISM4 boundary conditions. White numbers in the bottom represent the sea-level rise with respect to the PD state.

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
