# Peer review of "Antarctic Tipping points triggered by the mid-Pliocene warm climate"

_Climate of the Past, 2023_

## Author Comment (AC1)

Blue color: reviewer comment

Dark color: Author response

=====================================================================

**Summary**

The manuscript presents a study into 'tipping points' in the Antarctic ice-sheet system: climatic thresholds that, if crossed even briefly, lead to irreversible, large-scale retreat. Determining the existence and quantifying the thresholds of these tipping points can help understand the future evolution of the ice sheet. The authors study these tipping points from a palaeoclimate perspective. They have set up an ensemble of different realisations of the ice-sheet model Yelmo, and forced those realisations with output from a number of different climate models that have simulated the climate of the mid-Pliocene Warm Period. As that period represents the most recent time in Earth history that any significant retreat of the East Antarctic ice sheet might have occurred, it could help provide insight in the mechanics of these tipping points. The authors show that, depending on the amount of oceanic warming and the changes in precipitation predicted by the climate models, different sectors of the Antarctic ice sheet could have collapsed during this period. From this, they estimate the warming thresholds that represent the tipping points.

We thank the reviewer for the thoughtful comments and valuable suggestions, which will contribute to improving the quality of our manuscript. Below you can find our response to each comment.

**General comments**

In general, I think the manuscript does a good job of presenting the work. The introduction provides a good context, the methodology generally gives a good picture of exactly how the work was executed, and the results are presented clearly.

However, aside from a number of small, technical comments (outlined below), I have a few more significant concerns with the adopted methodology.

1) Initialisation. The ice-sheet model in this study is initialised by a steady-state spin-up, with no inversion procedure for basal friction, viscosity, or basal melt. The consensus that has emerged in the last few years is that the biases this introduces into the dynamic response of the model to a climate forcing, are no longer acceptable. The (currently stated) aim of this study is to quantify warming threshold beyond which certain sectors of the ice sheet will collapse. I believe the current initialization strategy introduces major errors in these numbers. Either the initialization strategy needs to be revised (which I think should become standard practice for all palaeoglaciological applications, just as it already is for future projections), or the aims of the manuscript need to be changed.

We understand the reviewer's concerns about our initialisation procedure.The reason we have chosen not to use this method here is that friction coefficients from past periods may not be the same as for the present-day (PD). Since the Antarctic Ice Sheet (AIS) has

undergone several periods of advance and retreat since the mid-Piacenzian warm period (mPWP) to PD, we could expect that due to erosion, friction coefficients are different.

The friction law used in this study follows

$$\tau_b = c_b \left( \frac{|\mathbf{u}_b|}{|\mathbf{u}_b| + u_0} \right)^q \frac{\mathbf{u}_b}{|\mathbf{u}_b|}$$

$$c_b = c_f \lambda N$$

$$\lambda = \begin{cases} 1 & \text{if } z_b \geq 0 \\ \max \left[ \exp \left( -\frac{|z_b|}{400} \right), 10^{-4} \right] & \text{if } z_b < 0 \end{cases}$$

where $c_f$ is a unitless coefficient (which in our study ranges from 0.1-1.0). Our simulations produce good results in terms of RMSE of ice thickness and surface velocities comparable to those of other groups in the context of ISMIP6 (Figure 1). Furthermore, there is no a priori reason to believe that optimized friction coefficients for PD would have been the same for the mPWP. Our approach has the benefit that basal friction adapts to changes in ice thickness and effective pressure as a result of the changes of the boundary condition of the mPWP with respect to the present day. Therefore, we believe that for the sake of our study, it is more beneficial to use a simple parameterization as in other paleo-studies (Quiquet et al., 2021), rather than optimized friction coefficients. Another future idea could be to include an active sediment mask to account for changes in erosion.

We will make it clearer in the revised manuscript.

2) Climate forcing. The method presented here, where the ice sheet is directly subjected to unchanging output from a GCM, is outdated. Even if setting up a coupled GCM-ISM is judged to be too expensive (which is probably the case for the multimillennial simulations presented here), there have been several papers in the last few years that describe more elaborate ways of forcing an ice-sheet model with pre-calculated GCM output. This way, the effects of the changing ice-sheet geometry on the climate, and the way these in turn feed back on the ice geometry, can be captured more accurately. Since the study involves major change in ice geometry (including the complete collapse of the West Antarctic ice sheet), these f+eedbacks should not be ignored.

Our lapse-rate factor includes changes in geometry. On one hand, PD climatologies are scaled with the surface elevation from RACMO2.3. On the other hand, we also take into account the surface elevation from the mPWP and pre-Industrial period provided by the PlioMIP2 protocol. This point will be made clearer in the next submission since it has raised questions about the methodology and the influence of topographic changes.

The other reviewer also had concerns about potential feedback mechanisms which may be neglected due to not coupling our ice-sheet model with an AOGCM. We have included the following discussion paragraph:

"Our forcing strategy based on an anomaly-snapshot method (i.e. one constant climatic snapshot from each AOGCM) ignores certain climate-interactions that could be relevant to the system. We take into account the surface melt-elevation feedback, by employing an atmospheric lapse-rate factor, and the albedo-melt feedback, which is considered within our ITM parameterisation. Nonetheless, probably one of the most important feedbacks not considered here is the effect of freshwater flux release from the AIS into the Southern Ocean. Results from Sadai et al. (2020) show that accounting for Antarctic ice discharges increases Southern Ocean temperatures, whereas in Bintanja et al. (2015) ice-shelf melt leads to a cooling of the Southern Ocean and an expansion of sea ice area. This points to the need for a more profound understanding of ice-ocean related processes within models.

A more sophisticated approach would include direct coupling between an AOGCM and our ice-sheet model. However, besides more computational resources, this would require constraints not only on our ice-sheet model parameters, but also on those of the AOGCM. The work of Berends et al. (2019) is a good example of an intermediate strategy, based on a matrix method. However, for this it is necessary to account for an AOGCM to produce several snapshots with varying $CO_2$ concentrations and ice-sheet coverage. In addition, running long transient experiments with this method still needs to be traded off with a lower ice-sheet resolution. In Berends et al., (2019), the AIS is simulated at a 40 km resolution. This is a potential explanation why they do not simulate a retreat in the East Antarctic region. Here we aim to obtain a more profound understanding of processes related to ice dynamics in part through a higher spatial resolution (16 km)."

3) Tipping points in dynamical systems. There have been a number of studies into tipping points from a dynamical systems point of view. These show, for example, that it matters for how long a threshold is crossed, or by how much, before a system tips. The steady-state approach adopted here excludes these temporal effects. For example, recently Stap et al. (2022, The Cryosphere) demonstrated that there are significant differences between the equilibrium response of the ice sheet (which is what is studied here), and the transient response (which is what occurs in reality). While they studied this in the even warmer Miocene, the conclusions apply to this study as well. The authors already mention that, in several of their simulations, the collapse of an ice-sheet sector can occur several thousand years after the onset of the climate forcing. Given the significant climate variability during the mid-Pliocene Warm Period, it is possible that this study significantly overestimates the ice-sheet retreat and the sea-level high stands during this period.

Indeed, we are applying a constant temperature for a long time period which may not mimic reality. Furthermore, we assume a constant steady-state whereas the transient behavior of the AIS may not respond equally. Thus, probably we could be overestimating the ice-sheet retreat and sea-level contribution since we forced with a warm period for a long model-time. Nonetheless, we applied a similar methodology as other ice-sheet modeling studies for the mPWP (deConto et al., 2021; Dolan et al., 2018; deConto and Pollard, 2017; Golledge et al., 2017; Yan et al., 2016; deBoer et al., 2015). We extended the discussion section as follows:

"In our study, the transient character of the climate system was neglected for the sake of simplicity and the poor knowledge on the transient forcing as well. Instead, we decided to force towards a steady mPWP state for an ensemble large enough to be statistically significant (more than 30 simulations) for 12 different mPWP conditions. This approach

permits us to assess Antarctic tipping points starting from PD conditions as well as the impact of the uncertainty associated with state-of-the-artc equilibrium mPWMP climatic conditions. This experimental setup goes in line with other studies, allowing for a similar comparison (Yan et al., 2016; DeConto and Pollard, 2016; DeConto et al., 2021). However, assuming a constant warming may lead to overestimation of sea-level contributions. As shown in Stap et al. (2022), the simulated Antarctic ice-volume evolution for the Miocene period reduces when the forcing is transient, rather than static. To our knowledge, only one study has simulated the transient evolution of the AIS under the Pliocene. The climate forcing in the transient evolution of Berends et al. (2019) did not reach the necessary conditions to lead to a retreat in the Wilkes basin, and thus simulated a lower sea-level contribution (Fig. S4c).

It is important to mention that exceeding a tipping point does not mean that the ice sheet will collapse immediately, but rather that it has reached the threshold temperature by which a retreat will be induced and further amplified by MISI. By plotting the one dimensional evolution of the WAIS (Fig. S2), we observe that the WAIS collapse usually occurs with a lag of 1000-5000 years from the application of the forcing. In some cases it can reach up to 25000 years. MISI is not only a matter of the oceanic temperature threshold, but also depends on the grounding-line position and the thermal forcing at this location, as well as precipitation. Thus, a transient character in the forcing could avoid certain ice collapses if the warming is not sufficiently long. Other factors, such as ice dynamics, could also delay (or accelerate) the grounding-line position reaching a pronounced retrograde bedrock that leads to a full collapse of the WAIS or other marine basins."

**Specific comments**

**L12:**
"…a higher-order ice-sheet model." DIVA is not a higher-order flow model. As Goldberg (2011) states, it is "a depth-integrated approximation to a higher-order flow model". Please change this here and throughout the manuscript.
**L106:**
"…similar to other 3D higher-order models." Same as before.

This raises an interesting question about model hierarchy. We followed the description of Lipscomb et al., (2019) which distinguishes between: (a) SIA and SSA approximation, (b) DIVA: depth-integrated higher-order approximation, (c)  3-D higher-order approximation based on Blatter-Pattyn (BP). As Lipscomb et al., (2019) we believe that DIVA constitutes a major improvement compared to hybrid-models and this should be reflected in the manuscript.

Schoof and Hindmarsh (2010) developed a depth-integrated model (see also "L1L2" model in Hindmarsh, 1993) highlighting the peculiarity that velocities can be computed to order $O(\varepsilon^2)$ with a normal stress that is truncated at order $O(\varepsilon)$ (see Eq. 3.32 in Schoof and Hindmarsh, 2010). Therefore, a second-order accurate velocity solution can be obtained only by keeping first-order terms in the normal stress. In our manuscript, we have changed the text  from "higher-order model"  to "depth-integrated higher-order model" following Lipscomb et al. (2019), but we believe it is accurate to explicitly state that the DIVA solver is second order accurate in the velocity solution.

L15: "…related to initial topography…" Do you mean initial ice thickness, or bed topography?

Ice thickness. We changed it in the manuscript.

L20: "…the likelihood of crossing them under future emission scenarios". You have not done any simulations involving future scenarios though, so I think this is phrased too strongly here.

We removed the last sentence to avoid misinterpretation.

L22: "ocean expansion" change to "thermal expansion".

Done.

L30-31: what is/are the source(s) for the different temperature thresholds you state here?

We included the references.

L39: "The thinning of ice shelves…" you mention hydrofracturing as a process causing this, but hydrofracturing causes shelves to disintegrate, not to thin. Choose a more appropriate phrasing.

We changed it to "The stability of ice shelves depends on several processes, such as…"

L48: Is there no more recent data on Pliocene CO2 concentrations than Haywood 2016?

We added two new references.

L113: This needs some more explanation. Provide the expressions for lambda and N, and explain what you mean when you say that "we will use it [c_f] for calibration of the model".

In the next submission we will add details on $\lambda$, $N_{eff}$ and $c_f$.

L120-121: This is maybe a bit too cautious. Earlier work on sub-grid scaling of friction (Feldmann et al., 2014; Leguy et al., 2021; Berends et al., 2022), has quite clearly shown that you can get good results at much coarser resolutions than the <100 m numbers mentioned in the MISMIP papers.

We included those references in that section to point out that sub-grid scaling of friction leads to good results at coarser resolution.

L123-125: The last word has not yet been said on the problem of sub-shelf melt at the grounding line. It is not automatically so that the NMP scheme is the best choice at coarse resolutions (see e.g. Berends et al., 2023, Journal of Glaciology), so I'd like to see a bit more discussion on how this affects your results.

Indeed, applying melting at the grounding-line would very likely have a big impact on the results. We have extended the discussion as follows:

"Another source of uncertainty is the melting at the grounding line. Observations have established that the ocean-induced basal melting is highest close to the grounding line and

decreases towards the ice-shelf front (Adusumilli et al., 2020). Ice-sheet models use different approaches which are typically no ocean-induced melting or partially ocean-induced melting (Seroussi and Morlighem, 2018; Leguy et al., 2021). In many coarse resolution ice-sheet models (more than 2 km resolution at the grounding line), no melting is applied directly at the grounding line since it can lead to overestimation of sub-shelf melting (Seroussi and Morlighem, 2018). Nonetheless, the work of Berends et al. (2023) demonstrated that applying melting at the grounding line (via a flotation criterion) in their model resulted in more appropriate for some settings. This could suggest that our results correspond to a lower limit since no melting is applied at the grounding line in our experiments. We expect that by adding melting at the grounding line, the collapse of the Wilkes basin would have been more likely for those AOGCM climates with higher oceanic thermal forcing. Basal melting parameterisations remain a source of uncertainty which need further investigation."

L133: "lambda_srf and c are parameters used to calibrate the AIS" What do you mean by this? Do you calibrate to achieve an SMB similar to observations/regional climate models when forced with reanalysis climate? Or do you calibrate to produce a steady-state ice sheet similar to observed? Both could be seen as appropriate for this work, but you should explain what you do and why.

We calibrated to PD ice thickness observations assumed at steady state. This point will be made clearer in the next submission.

**L140**:
More often, people adjust precipitation by ratios rather than differences (e.g. P = P_pd * (P_mod_Plio / P_mod_PD). Explain why you deviate from this practice.
**L237-243**:
This section is unclear. What do you mean when you say that the "precipitation anomaly lies around 85% of PD precipitation"? Do you mean that (P_mod_Plio - P_mod_PD) / P_PD = 0.85? Or that P_mod_Plio / P_mod_PD = 0.85? Or 1.85? The phrase "Thus, we see that a thermal forcing below 0.5 K can lead to a collapse of the WAIS if precipitation stays below 80% of PD" seems to suggest that the WAIS will collapse if the ocean is warmer than present, AND the precipitation is lower than present. Please rewrite this in a more understandable manner.
**L248-250**:
Again, your treatment of precipitation anomalies is very confusing. You state that a precipitation "three times more than PD rates" is similar to an anomaly of "around 130% of PD rates". I see no way for those numbers to be similar.

We do employ precipitation ratio anomalies, not differences. This was a typo we did not see and we thank the reviewer for pointing it out. We believe that this now clarifies all the other points  related to the treatment of precipitation. We will correct this typo in the next submission and it will be made clearer.

L142: There are more recent RACMO simulations of Antarctica than this.

Indeed, we could update our RACMO fields. Since we do not expect big changes with the latest RACMO version we went on with the previous dataset. Nonetheless, we will update to the latest RACMO version for future work.

L146-147: "…a lapse rate correction factor is applied…" What do you use as the reference surface elevation? RACMO uses a different elevation than the GCMs you used, how do you account for this difference?

To scale the PD climatologies we use the RACMO dataset (i.e., the surface elevation from RACMO2.3). Climatologies from the GCMs are scaled with the surface elevation provided by the PlioMIP2 protocol for the pre-industrial and the mPWP period. This will be made clearer in the manuscript.

L162: "PD fields are obtained from the ISMIP6 protocol" Technically, this protocol only dictates how to extrapolate any ocean temperature dataset into the sub-shelf cavities, and into the space currently occupied by ice or bedrock. I assume you mean you applied this protocol to the World Ocean Atlas dataset? If so, mention this.

Yes, we referred to the World Ocean Atlas dataset with sub-shelf temperatures extrapolated by the ISMIP6 protocol. We changed it in the manuscript.

**L162-163**:
"For computing the basal-melting rates at the mPWP, the Tf and So fields are changed with an anomaly method analogous to equation 4". The GCMs only provide temperatures for the open ocean. How do you compute the anomalies in the sub-shelf cavity? Do you apply the same ISMIP6 extrapolation protocol to the GCM data?
**L393-398**:
Ah, here is the detail I was missing earlier. This should already be explained in your methodology section.

We moved the paragraph to the methodology section.

L165: "For those cases, a spatially homogeneous temperature anomaly field of one fourth of the atmospheric anomaly was applied, following work by Golledge et al. (2015) and Taylor et al. (2012)." This seems rather ad-hoc. If you were to apply this method to the GCMs where you do have ocean data, how would the result compare to the actual GCM ocean temperatures?

Indeed, comparing with data from the other AOGCMs may provide more information here. The mean ratio between surface temperature and oceanic forcing for the AOGCMs here is 0.43, but it spans from 0.04 (MIROC4m) to 1.5 (NorESM1-F). Of course, depending on if the rate is higher than 0.25, then it would be likely to have more retreated marine basins, whereas if we took a smaller value we would expect a smaller sea-level contribution. We will maintain 0.25 (Golledge et al., 2015, Bulthuis et al., 2019) since it is based on other values from literature, and we would expect this uncertainty to lie within the inter-model uncertainty from the climatic forcing.

L169-170: "First we perform an ensemble of 180 ice-sheet simulations for the AIS with different dynamic configurations under steady PD climatic conditions using the ." Using the what? Oh, the suspense!

Fixed.

L177-180: How do your initialized geometries compare to the present-day in terms of root-mean-square errors in thickness and velocity? Having an approximately correct volume is a nice start, but in theory this could also be achieved by collapsing the Wilkes basin but grounding the Ross.

Indeed, we added to the SM the following plots of the simulated RMSE of ice thickness and velocity together with a spatial map of the mean. The 2D ice thickness and velocities both show reasonable distributions.

[Figure]

*Figure 1: Bar chart of the simulated PD RMSE in (a) ice thickness and (b) surface velocity for every simulation.*

[Figure]

*Figure 2: Mean PD state of all the PD simulations. (a) surface elevation (grey colors) and ice shelf thickness (orange); (b) surface velocity; (c) ice thickness (d) surface velocity anomalies with PD observations and its respective RMSE.*

L197: "Ice extension ranges from 9.2 times 10^6 km2" Why spell out the word "times"? I've not seen this notation anywhere else.

We changed it to "x"

L215: "The ice thickness is practically always negative" I assume you mean the thickness anomaly.

Yes, fixed. Thank you!

L222-224: "This spread in the EAIS and more specifically in the Wilkes basin points to an important role of the applied boundary conditions in the model response." What boundary conditions do you mean?

We changed it to

"This spread in the EAIS and more specifically in the Wilkes basin points to an important role of the ice dynamics."

L228: "Since an increase in oceanic forcing is thought to be the main driver of MISI…" Thought by whom? Also, consider a different phrasing than "driver" – by definition, an instability will result in a retreat that continues even in the absence of continued forcing, which only needs to act as a trigger.

Indeed, trigger is more appropriate in this context. We use oceanic forcing as a metric since it has a strong effect on the stability of ice shelves, which in turn reduces buttressing effect (Fürst et al., 2016). Now it reads:

"We find that oceanic forcing is the main forcing which defines the ice extent of marine basins (Figure 8)."

L229: "the ice extension" I think you mean ice extent, here and in other places throughout the manuscript.

Corrected.

L231: "this model result does not show realism in terms of sea-level equivalent" What do you mean by this?

We mean that this is the only AOGCM that simulates a negative sea-level contribution, which strongly disagrees with sea-level reconstructions and other proxy data. This will be made clearer in the next submission.

L235-236: "we focus on the four models that do not exceed 1 degree of oceanic anomaly" How do you define this anomaly? Global, Southern Ocean, Amundsen Sea, sub-shelf cavity? Sea surface or vertical average?

It is defined by basins (Figure S3) at grounding-line depth. This will be made clearer in the next submission.

L245: "we assume that thermal forcing is the main trigger" Do you mean oceanic or atmospheric thermal forcing?

We mean oceanic thermal forcing, this was made clearer in the manuscript.

L270-279: This section is unclear. Did you use a different initial ice thickness, or (also) a different bed topography? Did you use that as the initial state for your spin-up simulations, or only for your GCM-forced simulations? What do you mean by "the parameters from the ensemble that produced results closest to the mean value for every AOGCM forcing"?

We used the PD bedrock topography with the PRISM4 ice thickness. Instead of running the whole ensemble, we took for every AOGCM the dynamics parameters ($c_f$, $E_f$ and q) which simulated the closest value to the mean.

L305-307: This work by Richards et al. (2022) really should already be mentioned in the introduction.

Indeed, and we are grateful for this suggestion. We included a new paragraph in the introduction which reads as follows:

"Another approach to infer sea-level estimates from a modeling perspective is through Geodynamic Models. These models use glacial isostatic adjustment and mantle dynamic topography to compute and correct for sea-level records. An advantage is that they account for potential rebound effects which are difficult to assess on in-situ sea-level records. In the work from Hollyday et al. (2023) they used such a model to simulate the mantle flow from the Patagonian region. This resulted in lower mPWP sea-level estimates of 17.5±6.4 msle, and specifically an AIS contribution of 9.5±6.9 mSLE. Similar results are obtained in the work by Richards et al. (2022) where they simulate the Australian mantle deformation and compare it with proxy data from that region. They obtain a mPWP sea-level stand from 10.4-21.5 mSLE. Moucha and Ruetenik (2017) simulate a global sea-level contribution of 15 mSLE based on the US Atlantic shoreline. These studies reflect an overestimation in sea-level rise of in-situ records since lithospheric rebound is poorly considered."

L315: "this threshold is highly sensitive to structural dependence" What do you mean by "structural dependence"?

What we mean with structural dependence are model parameters or choices of parameterisations (like grounding line treatment) which have a great impact on the outcome and its application remains a matter of debate. We switched the term to "structural (i.e. related to the parameterisation choice) and parametric model dependence".

L304-316: This section contains a number of spelling/grammar errors.

This section will be corrected in the next submission.

L318: "This might seem counterintuitive…" It does. Your suggested explanation that "ice does not flow sufficiently fast to readvance again" contradicts the findings of the MISMIP experiments, which are founded on basic ice dynamics.

Indeed, it contradicts the findings. We believe it is related to the initialized PD state. We rephrased our text as follows:

"In our study we do not find a clear relation between ice dynamics and the simulated ice extent. Simulations forced with CESM1.0.5, simulate a slightly more retreated Totten basin for low enhancement factors (Fig. S5). We believe this is rather a consequence of the simulated PD state rather than ice dynamics, since we did not use any ice extent metric in the EAIS. In the MIROC4m model we find that a WAIS collapse is more likely to occur for high enhancement factors and low friction exponents, which promotes faster ice flow. In summary, although we observe some trends associated with the dynamic configuration for CESM1.0.5 and MIROC4m, no clear relationship can be found."

L323-324: "Such an analysis of structural dependence allows us to assess the sea-level uncertainties that arise from dynamical configuration and climatologies." Unclear what you mean by this.

We switched it to "Our analysis allows us to assess the sea-level uncertainties that arise from structural (model-related) uncertainties within one ice sheet model and climate-forcings from different AOGCMs "

L328-329: "Thus, a large ensemble parameter constraint like in our study, helps considerably to reduce uncertainty from ice-sheet models." No, you only used one ice-sheet model. Dolan et al. (2018) used three completely different models (ANICE, Sicopolis, and BASISM). It's not at all surprising that they find a larger spread.

Yes, our results refer to ice dynamics within one ice sheet model. We rephrase the paragraph to

"We find that the climatologies yield a larger uncertainty (~7 msle) than that resulting from the dynamic configuration if parameters are constrained with PD observations. Dolan et al. (2018) obtain more than 10 msle between different ice-sheet models, whereas we obtain less than 2 msle differences for simulations which are not close to tipping, and up to 5 msle differences for CESM1.0.5 due to the proximity of Wilkes basin to tipping or not(Error bars Fig. 5)."

L331-333: "Consistent with our results, ISMIP6 simulations forced with these and other climate models predict that Antarctic tipping points could be reached within this century" Your results do not support this at all. You have performed no future simulations.

The other reviewer also had concerns regarding this point. To avoid confusion we removed the paragraph since it is not clear how our results affect future projections.

L335-336: "Two of the models employed here (EC-Earth3.3, HadGEM3, see Table S1) belong to CMIP6 whereas the rest belong to CMIP5" What does that imply for your results?

 This paragraph was removed to avoid confusion.

L339-245: This section is vague, but I think it tries summarise one of my major concerns mentioned before: that there is a difference between equilibrium response and transient response, and that tipping points concern the latter rather than the former.

Yes, we will improve this section by adding a more detailed discussion of the differences that may arise between a transient forcing and a steady-state forcing, as discussed above.

L348: "an ensemble large enough to be statistically significant (more than 30 simulations)" How many simulations would you define as "statistically significant"? I find it hard to believe that you can put a number on this.

Statistically speaking, the more data you have, the more robust the conclusion is. In probability theory the distribution of a random sample converges to a normal distribution if the sample is large enough (the central limit theorem). Usually the number 30 is taken as a "thumbnail rule" to consider that the central limit theorem is applicable.

L352: "the transient evolution of Berends et al. (2019) allowed only for a WAIS collapse, avoiding other tipping points" Untrue. This model set-up "allowed" for ice-sheet collapse anywhere. That the climate forcing applied there did not result in a collapse is another matter.

Yes, the reviewer is correct. We rephrased it to

"The climate forcing applied in the transient runs of Berends et al., (2019) did not lead to a retreat in the Wilkes basin, and thus simulated a lower sea-level contribution (Fig. S4c)."

L357-358: "their results also show that starting from PRISM4 conditions leads to higher sea-level contributions and a less extended AIS during the mPWP. This result is expected" Those results were obtained with ice-sheet models that did not yet include any grounding-line treatment. Any hysteresis they found could well be a model artefact rather than a physically meaningful result.

Indeed, it could be an artifact, but smaller ice sheets also take into account the melt-elevation feedback. Now it reads:

"Their results also show that starting from PRISM4 conditions leads to higher sea-level contributions and a less extended AIS during the mPWP. This result is expected since a smaller ice sheet is forced by warmer surface temperatures due to the melt-elevation feedback, captured in our experiments through a lapse-rate factor. In addition, growing back on a retrograde marine basin needs a strong decrease in ocean temperature due to the hysteresis behavior of the ice sheet."

L358-360: the arguments after "the one hand" and "the other hand" seem to point in the same direction?

Yes, we changed it to avoid confusion.

L362-363: "before the mPWP, $CO_2$ concentrations were below the pre-Industrial period, with sea-level estimates also below PD" Do you mean the M2 cold excursion? Be specific.

Yes, we refer to M2. We have rephrased the text as follows:

"The mPWP was preceded by a large global glaciation during Marine Isotope Stage M2 ca. 3.3 Ma BP (Rohling et al., 2014; Stap et al., 2016). During that period, the AIS evolved towards a modern-like configuration (Berends et al., 2019)."

L366-367: "only 3 out of 12 AOGCM models can be considered to realistically simulate warm Pliocene conditions, according to our simulations" I do not think your results are strong enough to discredit the results of any of the GCMs. Phrase this less strongly.

Indeed, this sentence sounds too strong the way it is framed and we cannot conclude this from just one ice-sheet model. We rephrased it to

" Our model only simulates a retreat in the Wilkes basing, supported by reconstructions, for 3 out of 12 AOGCM models."

L373-383: Unclear where you are going with this. Comparing to the Abumip results is meaningless here, as in those simulations all floating ice is destroyed in the models. The flux condition is not something you use, nor something you need – Alex Robinson already showed in the original Yelmo paper that your sub-grid friction scaling scheme works just fine.

We removed the part of ABUMIP and left a discussion paragraph of grounding-line treatment in our ice-sheet model. Now it reads:

"As shown by Pattyn et al., (2013), high resolution is needed at the grounding line to simulate accurate grounding-line migrations. In order to overcome this, ice-sheet models use different techniques at the grounding line to compensate for coarse resolution, such as flux conditions (Schoof 2007, Tsai et al., 2015) or scaling friction at the grounding line with its grounded ice fraction. In our study we use the latter technique which has shown to simulate realistic grounding-line migrations on idealized domains (a thorough description is presented in  Robinson et al., 2020). We also ensure that effective pressure, which enters the basal friction equation, tends to zero as the ice thickness approaches flotation (Leguy et al., 2014). Nonetheless, grounding-line parameterisations remain as a source of uncertainty that can strongly influence the retreat of marine based glaciers prone to MISI"

L385: "the particular melting implementation at the grounding line is somewhat arbitrary" A strange way to phrase this. The fact that no perfect solution exists yet does not make the existing imperfect solutions "arbitrary".

We rephrased it to

"Ice-sheet models use different approaches which typically range from no ocean-induced melting to partially ocean-induced melting."

L391-392: "Given that we do not apply flux conditions or grounding-line melting, our results are more conservative than other studies" I'd argue that it is the other way round. Your model includes a sub-grid friction scaling scheme, which has been shown to work well. The models you compare with did not have any special grounding-line treatment, so that they likely would have severely underestimated any grounding-line retreat.

Yes, this is true, but since the focus of that paragraph is grounding-line melting and we do not apply grounding-line melting we believe that our results are rather conservative compared to other studies. We mention both effects in the discussion section.

L410-411: "Consequently, the model initialized with the PRISM4 ice-sheet thickness displayed persistent differences in simulated AIS characteristics compared to other initializations." That begs the question of how the ice sheet came to be so small in the first place, which I think needs to be discussed somewhere.

This raises an important question regarding PlioMIP2 boundary conditions. Direct evidence for ice free conditions are scarce for the Antarctic ice Sheet. Proxy records are limited to some marine records and land regions in the McMurdo Death valley (Shakun et al., 2018). PRISM4 boundary conditions are the same as PRISM3, and were generated using the British Antarctic Survey Ice Sheet Model (BASISM) with boundary conditions from an Atmospheric General Circulation Model (Dowsett et al., 2016).

Although this is not the aim of this study, we believe, based on our results and latest GIA model results (Richards et al., 2022; Hollyday et al., 2023), that the employed PlioMIP2 boundary conditions represent a lowest case scenario and that future work should potentially focus on the possibility of a larger Antarctic Ice Sheet during the mPWP. We added the following discussion paragraph:

"Since our Antarctic sea-level contributions do not exceed 10 mSLE, our simulations do not support a global sea-level contribution of more than 20 mSLE as suggested by some reconstructions (Dumitru et al., 2019; Hearty et al., 2020). Nonetheless, recent work done with Geodynamic Models suggest a lower contribution at the mPWP than proxy data. These models simulate dynamic topographic changes on specific domains, namely the Patagonian region (Hollyday et al., 2023), the Australian region (Richards et al., 2022) and the Atlantic shoreline (Moucha and Ruetenik, 2017). The main advantage compared to proxy data is that processes that are difficult to assess on in-situ measurements and have a big impact, such as geostatic uplift, can be considered. These results are then compared to proxy measurements from that region to assess the reality of their simulation. The new sea-level estimates reduce the global sea-level contribution significantly: 17.5 ± 6.4 msle (Hollyday et al., 2023); 16.0 ± 5.5 msle (Richards et al., 2022); 15 msle (Moucha and Ruetenik, 2017). Assuming that Greenland was almost fully melted (~ 7.4 msle, Morlighem et al. (2017)), with such a revised sea-level reconstruction, our results are inside the geological constraints if Wilkes basin collapses via high oceanic thermal forcing or with low precipitation rates, as in MRI-CGCM2.3 (Table 1 in SM). Richards et al. (2022) even go one step further and argue that the impact of the proposed MICI mechanism (DeConto and Pollard (2016; DeConto et al., 2021) is overestimated. Though this is not the scope of our work, these new results could highlight the need for new boundary conditions during the mPWP for the AOGCM, mainly a larger and thicker AIS than previously thought."

L417: "a lowering of PD precipitation could lead to such an irreversible retreat." Do you think a future decrease in precipitation is realistic?

This seems unrealistic since ISMIP6 projections show increased precipitation. We rephrased it to:

*"[...] ...a lowering of PD precipitation, although not projected by AOGCMs in Antarctica (Seroussi et al., 2020), could lead to an irreversible retreat. Still, in the long term, such an instability could be also caused by an increase in melting. "*

L419: "our simulated sea-level contributions ranged from -1.8 mSLE to -9.6 mSLE considering the whole ensemble." A negative contribution indicates a sea-level drop.

Indeed, we changed the sign.

L423: "as well as grounding-line migrations" Unclear what you mean here.

We changed it to "Our results reinforce the hypothesis …"

L424: "sea-level standings" = sea-level high stands?

Corrected.

**References:**

- Quiquet, A., Roche, D. M., Dumas, C., Bouttes, N., and Lhardy, F.: Climate and ice sheet evolutions from the last glacial maximum to the pre-industrial period with an ice-sheet–climate coupled model, Clim. Past, 17, 2179–2199, https://doi.org/10.5194/cp-17-2179-2021, 2021.
- R. C. A. Hindmarsh, Qualitative dynamics of marine ice sheets, Ice in the Climate System (ed. W. R. Peltier; Springer, Berlin 1993) 67–99.
- Lipscomb, W. H., et al.: Description and evaluation of the Community Ice Sheet Model (CISM) v2.1, Geosci. Model Dev., 12, 387–424, https://doi.org/10.5194/gmd-12-387-2019, 2019.
- Stap, L. B., et al.: Net effect of ice-sheet–atmosphere interactions reduces simulated transient Miocene Antarctic ice-sheet variability, The Cryosphere, 16, 1315–1332, https://doi.org/10.5194/tc-16-1315-2022, 2022.
- Bulthuis, K., et al.: Uncertainty quantification of the multi-centennial response of the Antarctic ice sheet to climate change, The Cryosphere, 13, 1349–1380, https://doi.org/10.5194/tc-13-1349-2019, 2019.
- Golledge, et al.: The multi-millennial Antarctic commitment to future sea-level rise. Nature, 526, 421–425, https://doi.org/10.1038/nature15706, 2015.
- *Shakun, J. D., Corbett, L. B., Bierman, P. R., Underwood, K., Rizzo, D. M., Zimmerman, S. R., Caffee, M. W., Naish, T., Golledge, N. R., and Hay, C. C.: Minimal East Antarctic Ice Sheet retreat onto land during the past eight million years, Nature, 558, 284–287, https://doi.org/s41586-018-0155-6, 2018.*
- *Fürst, J. J., Durand, G., Gillet-Chaulet, F., Tavard, L., Rankl, M., Braun, M., and Gagliardini, O.: The safety band of Antarctic ice shelves, Nature Climate Change, 6, 479–482, https://doi.org/10.1038/nclimate2912, 2016.*

---

## Author Comment (AC2)

Blue color: reviewer comment

Dark color: Author response

======================================================================

This is a review of the submission "Antarctic Tipping points triggered by the mid-Pliocene warm climate" of Blasco et al to Climate of the Past.

The paper is an interesting contribution to the body of work utilising the Pliocene Model Intercomparison Project Phase 2 simulations. It builds on a number of ice-sheet modelling experiments conducted for this period and takes some elements of its experimental design from previous ice sheet model Intercomparison efforts (e.g. PLISMIP). The paper is well written in terms of language and readability. The figures are clear and of high quality.

The study takes a perturbed parameter approach to considering the configuration of the Antarctic Ice Sheet (AIS) during the mid-Piacenzian. Results in the study are used to make inferences of potential tipping point thresholds that may have been passed in the mid-Piacenzian, leading to retreat of large sectors of the AIS. The study provides a step forward on the work of Dolan et al, and De Boer et al, who use a similar modelling framework for exploring uncertainty, in that it considers more thoroughly parameter uncertainty within one ice sheet model.

We thank the reviewer for the positive feedback and suggestions, which will contribute to improving the quality of our manuscript. Below you can find our response to each comment.

**General comments:**

The authors make good use of the data available through PlioMIP and provide a good overview of their results. My general feeling is that the results presented here present a pragmatic approach to understanding uncertainties in ice sheet predictions for this time period. I am less convinced that the overall model set up is sufficient to draw strong conclusions around Antarctic tipping points. Through necessity, the modelling framework needs to ignore certain climate-ice sheet feedbacks, but I think a more thorough assessment of these would be needed to draw robust conclusions on tipping points. It would be useful for the authors to elaborate further on the other ways of considering/modelling tipping points, and the limitations of their current study. This does not question the overall results of the study, which in my view are sound, but there is a reflection required on how these results are interpreted when considering future Antarctic ice sheet changes.

Indeed, by using only a stand-alone model, it is difficult to assess the role of climate-ice sheet interactions which could have an impact on tipping points of the Antarctic Ice Sheet (AIS). However, running coupled experiments requires overcoming several technical issues:

- The same way we needed to assess a range of unknown parameters which created a realistic present-day (PD) ice sheet, this has to be done also for the climate model.
- Even if the parameter space of the ice sheet and the climate models are constrained, once coupled, they can lead to different results than the reconstructions, which would require a new parameter constraint.

- Running coupled experiments needs large amounts of computing time and space.

The work of Berends et al., (2019) is a good example of a coupled model (HadCM3 with ANICE). However, in order to run these simulations, they needed to make a trade-off between a coupled model and ice-sheet resolution (40 km). This is a potential explanation why they do not simulate a retreat in the East Antarctic region. Here we aim to obtain a more profound understanding of processes related to ice dynamics in part through a higher spatial resolution (16 km).

We take into account some climate interactions, such as the surface melt-elevation feedback, by employing an atmospheric lapse-rate factor, and ice-albedo effects, which are considered within our ITM parameterisation. Nonetheless, probably one of the most important feedbacks which is not considered here would be the effect of freshwater flux (FWF) release from the AIS into the Southern Ocean (SO), which remains a matter of debate. Results from Sadai et al., (2020) show that accounting for Antarctic ice discharges increases SO temperatures. These results contradict those from Bintaja et al., (2017), where ice-shelf melt leads to a cooling of the SO and an expansion of sea ice area. This points to the need for a more profound understanding of ice-ocean related processes within models.

Regarding tipping points, we believe that our results are robust for the AIS. Tipping points are thresholds that, once reached, lead to positive-feedback mechanisms. Of course, ice sheet-climate interactions can hamper (or facilitate) reaching those thresholds. Assessing tipping points with a coupled ice-sheet-ocean model has not been done yet. Our results show that a 1 degree oceanic anomaly leads to a WAIS collapse and a 3 degree anomaly to a Wilkes collapse, consistent with other modelling results (McKay et al., 2022). We look forward to coupling an ice-sheet model together with an AOGCM in the future to improve our results and our understanding of the mPWP as well as Antarctic tipping-points, however, this is in the scope of future work. We have included the following discussion in the manuscript:

"Our forcing strategy based on an anomaly-snapshot method (i.e. one constant climatic snapshot from each AOGCM) ignores certain climate-interactions that could be relevant to the system. We take into account the surface melt-elevation feedback, by employing an atmospheric lapse-rate factor, and the albedo-melt feedback, which is considered within our ITM parameterisation. Nonetheless, probably one of the most important feedbacks not considered here is the effect of freshwater flux release from the AIS into the Southern Ocean. Results from Sadai et al. (2020) show that accounting for Antarctic ice discharges increases Southern Ocean temperatures, whereas in Bintanja et al. (2015) ice-shelf melt leads to a cooling of the Southern Ocean and an expansion of sea ice area. This points to the need for a more profound understanding of ice-ocean related processes within models.

A more sophisticated approach would include direct coupling between an AOGCM and our ice-sheet model. However, besides more computational resources, this would require constraints not only on our ice-sheet model parameters, but also on those of the AOGCM. The work of Berends et al. (2019) is a good example of a coupled model. However, in order to run these simulations, one trade off is a lower ice-sheet resolution (40 km). This is a potential explanation why they do not simulate a retreat in the East Antarctic region. Here we aim to obtain a more profound understanding of processes related to ice dynamics in part through a higher spatial resolution (16 km)."

The authors make very few comments about the comparison of their results for PlioMIP2 and those already published for PlioMIP1 in terms of any significant changes between the climatological forcing. This is possibly beyond the scope of this study, but it would be interesting to try and assess the impact of the changes to boundary conditions between PlioMIP1 and 2 and their effect on Antarctica. While the ice sheet boundary condition remained the same, others (e.g. topography, ocean gateways) did not. Are there any systematic changes within the climatologies that would have an impact on the spread of sea-level predictions here?

We agree with the reviewer's opinion that a comparison between PlioMIP1 and PlioMIP2 provides an interesting framework to understand how updated boundary conditions affect the AOGCM and the AIS, however this is indeed behind the scope of our study.

In the following figure (Figure A) from Haywood et al., (2021), they show the multi-model annual mean surface temperature and precipitation for PlioMIP1 and PlioMIP2 and its differences. The main outcome in the Antarctic region and Southern Ocean is that PlioMIP2 models simulate warmer temperatures and a similar precipitation pattern. Thus we would expect a smaller AIS than from PlioMIP1.

**Temperature anomaly:**

**a) PlioMIP2**

**b) PlioMIP1**

**c) PlioMIP2 - PlioMIP1**

**Precipitation anomaly:**

**d) PlioMIP2**

**e) PlioMIP1**

**f) PlioMIP2 - PlioMIP1**

[Figure]

Figure A from Haywood et al., (2021): *"(A) PlioMIP2 and (B) PlioMIP1 multi-model annual mean surface air temperature (SAT) differences (over land) and sea surface temperature (SST) differences (over oceans) in °C, compared to the pre-industrial era. (C) Difference between PlioMIP2 and PlioMIP1 multi-modal means (°C). (D) PlioMIP2 and (E) PlioMIP1 multi-model annual mean total precipitation rate (mm/day) differences (compared to the pre-industrial era). (F) Difference between PlioMIP2 and PlioMIP1 multi-modal means (mm/day). Circles represent proxy-derived SST and SAT anomalies in (A) from McClymont et al. (2020) and Salzmann et al. (2013) respectively. Proxy-derived SST and SAT anomalies in (B) from Dowsett et al. (2010) and Salzmann et al. (2013) respectively."*

Finally, it would be beneficial for the authors to provide a brief summary of non-sea level related records that could help assess whether or not ice-retreat in certain areas would be expected during the mPWP.

Indeed we focused mainly on sea-level related records in the manuscript, as we did not find an extensive bibliography related to non-sea level related records. We hope that the following updated paragraph in the introduction provides clarifications on the expected retreat of Wilkes basin during mPWP :

"One key question in Antarctic reconstructions and simulations is whether the Wilkes Basin retreated or not during the mPWP (Wilkes basin illustrated in Fig. 8 and Fig. S3). Today, the WAIS and the Greenland Ice Sheet (GrIS) sum up to make a total of 10 mSLE (Morlighem et al., 2017, 2020). Thus, in order to achieve a sea-level rise far beyond 10 mSLE, a significant response in the EAIS is required. Marine records close to the Wilkes basin reinforce the hypothesis of such a retreat. Deposition of ice- rafted debris close to the Wilkes basin shows enhanced iceberg activity during the mPWP (Patterson et al., 2014; Bertram et al., 2018). This can be interpreted as a consequence of ice-sheet retreat with its related calving events. In addition, land-based sediment records of the EAIS show low concentrations of cosmogenic isotopes, which indicate that land-based regions experienced minimal retreat during the mPWP (Shakun et al., 2018). This points to a response of marine-based regions to explain high sea-level records"

**Specific comments:**

L4: the use of mid-Pliocene Warm Period is no longer the most up-to-date way of referring to this time interval (see the note on terminology in Dowsett et al 2023, Stratigraphy). Please also refer to Haywood et al (2020) for the definition of the time period used for PlioMIP2 - mid-Piacenzian warm period (3.264 to 3.025 million years ago).

Thank you for this clarification. We will update the terminology in the next submission of the manuscript.

L37: Such rather than Suc

Corrected. Thank you!

L50: Sea-level estimates noted here are inconsistent with those in the abstract.

We changed it for consistency.

L170: Sentence missing ending

Fixed.

L185: Do all simulations used or tried reach the steady-state condition required? It would be useful to note this as it is unclear from Figure S2 - were any excluded if they did not reach a steady-state?

Yes, our PD spin-up of $10^6$ years is sufficiently long to ensure that our ice sheet reaches a steady present-day state as shown in the following Figure 1. This point will be made clearer in the manuscript.

[Figure]

*Figure 1: Temporal evolution of the (a) ice-volume difference in terms of msle with respect to PD; (b) grounded-ice extent under PD climatological conditions. Blue lines represent simulations that differ less than 1 mSLE and 2.5 10⁶ km² with observations (a deviation of 2% from observed values) and are considered for mPWP simulations.*

L195 onwards: use X rather than times.

Changed.

L214: perhaps use 'assess the spatial origin'

Done.

L246: can you clarify what defines 'high probability' in this context?

We have specified now with numbers.

L274: missing end bracket

Fixed.

L325: Comparison with Dolan et al is useful, but it is unclear from the way it is written if the authors infer that they agree or disagree with the overall conclusion of a high climate model dependency of AIS reconstructions. Please consider alternative phrasing. The quoted sea level figures for 'simulations which are not close to tip' needs further elaboration.

The original text read as follows:

"Contrary to Dolan et al. (2018), we find that the climatologies yield a higher uncertainty (~7 msle) than the dynamical configuration, if parameters are constrained with PD observations. Dolan et al. (2018) obtain more than 10 msle between different ice-sheet models, whereas we obtain less than 2 msle differences for simulations which are not close to tip, and up to 5 msle differences for CESM1.0.5 due to its proximity to tip or not in the Wilkes basin (Error bars Fig. 5). Thus, a large ensemble parameter constraint like in our study, helps considerably to reduce uncertainty from ice-sheet models"

We changed it to:

"Dolan et al., (2018) find that climatic uncertainty leads to more than 10 msle. Our results yield a smaller uncertainty (~7 msle). However, the climatic outputs used by Dolan et al., (2018) belonged to the PlioMIP1 experimental setup whereas ours belonged to the PlioMIP2 experimental setup. Ice-sheet models uncertainty leads to more than 10 msle in Dolan et al., (2018) and deBoer et al., (2015). Our results suggest that the uncertainty associated with ice dynamics within an ice-sheet model is less than 2 msle for the majority of AOGCMs. Nonetheless, we find up to 5 msle differences for CESM1.0.5, where different dynamic configurations can lead to a retreat of the Wilkes basin (error bars in Fig. 5)."

L331: the statement that ISMIP6 simulations around tipping points over the next century being consistent with the results presented here seems a little ungrounded, given that these results are for a steady state following thousands of years.

Indeed, since it can lead to misinterpretation the paragraph was removed from the discussion section.

L364: Regarding the note around cooler conditions prevailing before the mPWP, it would be useful to provide some further context on this. Do you mean the MIS M2 event? Throughout this period there will have been orbital forcing that is not taken into account here.

Yes, we refer to the MIS M2 event (~ 3.3 Ma BP). We wanted to highlight that during that time cooler conditions prevailed allowing the ice sheet to extend to modern-like configurations (Berends et al., 2019). We have rephrased the paragraph as follows:

"The mPWP was preceded by a large global glaciation during Marine Isotope Stage M2 ca. 3.3 Ma BP (Rohling et al., 2014; Stap et al., 2016). During that period, the AIS evolved towards a modern-like configuration (Berends et al., 2019). Therefore, starting from PD initial conditions can help to assess the realism of the simulated mPWP from the AOGCMs. For instance, if the retreat of Wilkes basin is a necessary condition for an accurate mPWP representation, then only 3 out of 12 AOGCM models can be considered to realistically simulate warm Pliocene conditions, according to our simulations."

L417: use of "irreversible retreat" terminology should perhaps be reconsidered here in reference to the Pliocene, as WAIS has clearly advanced since then. However, I am not clear if this sentence does refer to the mPWP or PD – it would benefit from rephrasing I think.

We rephrased it to "[...] and that even a lowering of PD precipitation could lead to a potential retreat of marine basins" to avoid any confusion.

L419: Is this Pliocene only here, and does it include both ice-sheet initial conditions?

No, it is only starting from PD conditions. We therefore rephrased it to:

"Finally, our simulated sea-level contributions ranged from 1.8 mSLE to 9.6 mSLE considering the whole ensemble starting from PD conditions, and 15.5 mSLE to 25.6 mSLE when starting from PRISM4 conditions."

Fig 6: Just for clarity, does 100% probability mean that all simulations agreed? Do any of the results presented reach that 100% mark, or are they mostly between 80% and 100%. It could be useful to have a distinction in colour for when they do all agree.

Yes, 100% means all the simulations agree. We changed the figure and colorbar accordingly.

[Figure]

*Figure 6: Ice-collapsed probability of the ensemble for every AOGCM. Darkred/Blue colors indicate a high/low probability of a collapsed region. Gray colors show grounded ice for all the ensemble simulations.*

Table S1: additional comma in caption

Done.

**References:**

- *Berends, C. J., et al.: Modelling ice sheet evolution and atmospheric $CO_2$ during the Late Pliocene, Clim. Past, https://doi.org/10.5194/cp-15-1603-2019, 2019.*
- *Sadai, S., et al.: Future climate response to Antarctic Ice Sheet melt caused by anthropogenic warming, Sci. Adv., https://do.org/10.1126/sciadv.aaz1169, 2020.*
- *Bintanja, R., et al.: The effect of increased fresh water from Antarctic ice shelves on future trends in Antarctic sea ice, Annals of Glaciology, https://doi.org/10.3189/2015AoG69A001, 2017.*
- *McKay, D. A., et al.: Exceeding 1.5°C global warming could trigger multiple climate tipping points, Science, https://doi.org/10.3189/10.1126/science.abn7950, 2022.*
- *Haywood, A. M., et al.: PlioMIP: The Pliocene Model Intercomparison Project, Past Global Changes Magazine, https://doi.org/10.22498/pages.29.2.92, 2021.*
- *Shakun, J. D., Corbett, L. B., Bierman, P. R., Underwood, K., Rizzo, D. M., Zimmerman, S. R., Caffee, M. W., Naish, T., Golledge, N. R., and Hay, C. C.: Minimal East Antarctic Ice Sheet retreat onto land during the past eight million years, Nature, 558, 284–287, https://doi.org/s41586-018-0155-6, 2018.*

---

## Author Response (AR1)

Blue color: Editor's comment
Black color: Authors response

=====================================================================

**Initialization procedure:**

While both reviewers are to varying extents overall favourable, they have both raised issues that need to be addressed in the revisions. Tijn Berends has raised three major concerns about initialization and basal drag, tipping points under steady state forcing, and climate forcing which have been only partly addressed in the author responses. I urge some additional experiments to better address the concerns raised, where feasible.

We thank the editor for his comments. Concerning initialization, we did not use optimized friction coefficients because we believe this approach is not entirely valid for the period investigated. Given the long time elapsed since the Pliocene, we cannot assume constant basal conditions, as both the ice geometry and the temperature varied considerably with respect to the present day (PD). Nevertheless, we have followed the reviewers' suggestion and performed a new set of experiments where we have optimized friction coefficients and basal-melting rates following Lipscomb et al., (2019) for the same set of values of enhancement factors and sliding exponents that simulated realistic PD states in our first submission. First we optimize for 30.000 years for PD conditions. The ensemble mean simulated PD state is shown in Figure 1, together with the root-mean square errors. Figure 2a and 2b show the root-mean square error of each simulation and Fig. 2c the ice volume and extent difference with the observations.

[Figure]

[Figure]

Figure 1: Mean PD state of all the optimized PD simulations. (a) surface elevation (grey colors) and ice shelf thickness (orange); (b) surface velocity; (c) ice thickness (d) surface velocity anomalies with PD observations and its respective RMSE.

[Figure]

Figure 2: RMSE of ice thickness (a) and surface velocity (b) of the ensemble of simulations performed with the same set of values that simulated realistic PD present-day states in our first submission. (c) Ice volume and extent difference with the observations.

We then force these PD states with the PlioMIP2 climatologies (Figure 3). In terms of sea-level contribution, the optimized results are similar to those of the simulations with non-optimized friction coefficients, but the spread is lower, since we consider less simulations (9 optimized simulations, 31 non-optimized simulations). In terms of ice extension, both cases show similar values except for MIROC4m, which shows a lower ice extent. However, this is not surprising at all, since the optimized simulations have not reached equilibrium. If we let the optimized experiments run for 30.000 years with PD forcing we see that the ice volume decreases for 7 of 9 cases indicating a WAIS collapse (Figure 4). Such a trend in ice volume for optimized friction coefficients has been observed in other ice-sheet models for even shorter timescales (i.e. *Seroussi et al., in prep.; Coulon et al., 2023*). Though our optimized values simulate similar Pliocene contributions as our non-optimized case, we believe that basal-friction coefficient optimization is not the best approach for long timescales since it leads to a trend. Since, in addition, this approach is not totally justified, we prefer to maintain our original methodology.

[Figure]

Figure 3: Boxplot of the simulated (a) sea-level contribution (positive/negative numbers indicate a lower/higher ice volume); (b) grounded ice extent for every AOGCM. The scatter-point shows the mean values of the ensemble. The error bars represent the lowest/highest simulated AIS state starting from PD conditions. Light shaded colors at the right show the sea-level uncertainty ranges from the studies of deBoer et al., (2015, brown); Yan et al., (2016, pink); Golledge et al., (2017, red); DeConto and Pollard (2017, blue); Dolan et al., (2018, green); Berends et al., (2019; purple); DeConto et al., (2021, orange); Richards et al., (2022; yellow); Hollyday et al., (2023; grey). The dashed black line in (b) represents the PD grounded ice extent.

[Figure]

*Figure 4: Ice volume evolution and grounded ice extent for PD optimized experiments under PD forcing.*

**Technical comments:**

20 terrain-following vertical layers.
**do you mean sigma coordinates?**

Yes, we have changed it in the updated manuscript. .

basal friction is scaled at the grounding-line points with its proportional grounded fraction
**incomprehensible**

Indeed; in the updated manuscript it is now clearer.

**eq 4 What exactly to do you mean by Tatm? 2 meter air temperature or ?**

Indeed, the 2 m air temperature. It  is made clearer in the following submission.

order to account for surface temperature and precipitation changes in elevation, due to the Clausius-Clapeyron relation, a lapse rate correction factor is applied, 0.008 K m−1 for annual

temperatures and 0.0065 K m−1 for summer temperatures (Ritz et al., 1996; De- Conto and Pollard, 2016; Quiquet et al., 2018; Albrecht et al., 2020).
**eq 4 What exactly to do you mean by Tatm? 2 meter air temperature or ?**
**not clear how lapse rate is used for precipitation changes also, make clear as to whether the indicated lapse rates are based on free air vertical gradients, present-day surface elevation gradients, or 2 m air temperature gradients. And justify your choice if not based on the latter.**

Equations 4 and 5 refer to sea-surface elevation. PD climatologies  obtained from RACMO2.3 as well as from the mPWP are scaled to sea-surface elevation through a lapse-rate factor with the surface elevation provided by RACMO2.3 and PlioMIP2, respectively. The lapse rate factor is taken as a free parameter with the typical value of -0.008 K/m for annual temperatures and to -0.0065 K/m for summer temperatures. Then, these climatologies are scaled to the ice-sheet surface elevation through the same lapse-rate factor. This ensures that changes in elevation are taken into account in our simulations and any potential bias driven by elevation differences are avoided. The lapse rate factor affects precipitation via an exponential temperature scaling - this is now clarified in  the new manuscript.

**model description far from complete. Missing (need not be extensive) descriptions: GIA, bed thermal model, surface melt refreezing, calving, ..**

If the temperature at the ice base reaches the pressure melting point, then it is set to the pressure melting point, and the basal mass balance is diagnosed as in Cuffey and Paterson (2010), where the geothermal heat flow field is obtained from Davies (2013). The glacial isostatic adjustment is computed with the elastic lithosphere-relaxed asthenosphere method (Le Meur and Huybrechts, 1996), where the relaxation time of the asthenosphere is set to 3000 years.The calving rate C is derived as a sum between the principal stresses (τ1 and τ2) as in Lipscomb et al. (2019). From the computed melting through the ITM model we assume that a 60% refreeze as in Robinson et al. (2010). All of these details have now been included in the new manuscript.

Fig S1:
**are the values plotted total or grounded fraction?**

Only for grounded ice. We changed the y-axis description.

**there are a few place where you use ice "extension", when you mean 'area', eg Fig 5 caption, Fig S1**

We changed it to "extent".

**Ocean forcing : make clear if you account for subglacial meltwater discharge on ocean salinity, and justify if this is not the case.**
**what about uncertainties in marine climate/circulation which controls submarine ice melt?**

We do not take subglacial meltwater discharge into account since we do not couple any AOGCM to our ice sheet model. The updated manuscript includes a discussion on this.

**do you really mean "probability" without doing a proper statistical analysis?**
**please provide a clear argument why this would be "statistically significant"? I don't see one**

Statistically speaking, the more data you have, the more robust the conclusion is. In probability theory the distribution of the mean of a random sample converges to a normal distribution if the sample is large enough (the central limit theorem). Usually 30 and above are considered as valid sample sizes to apply the central limit theorem. We give our simulations which fall inside our threshold range (< 2% difference with PD ice volume and extension) the same weight. Then, we use boxplots to assess the 1st quartile (25%) and 3rd quartile (75%). Our updated uncertainty range in the manuscript is included inside the interquartile range. We believe this can be interpreted as a statistical analysis.

**some uncertainties not addressed: earth rheology for GIA, geothermal heat flux, climate downscaling Crow et al, ( in this same special edition).**

Indeed, we have extended our discussion section by mentioning uncertainties which were not previously considered in our study, such as GIA, geothermal heat flow, a spatial variable lapse-rate factor or coupling with an AOGCM.

*"Here, cf is a dimensionless field representing the basal properties of the base, such as soft/hard beds. Here we will use it for calibration of the model"*
**you need to document somewhere how this is done. And by calibration, do you just mean tuning?**
**need more details about parameter selection. Eg, is PD ice volume and area only sieve conditions? What about fit to observed grounding lines?**

Yes, with calibration we mean tuning towards PD conditions.

The friction law used in this study follows

$$\tau_b = c_b \left( \frac{|\mathbf{u_b}|}{|\mathbf{u_b}| + u_0} \right)^q \frac{\mathbf{u_b}}{|\mathbf{u_b}|}$$

$$c_b = c_f \lambda N$$

$$\lambda = \begin{cases} 1 & \text{if } z_b \geq 0 \\ \max\left[\exp\left(-\frac{|z_b|}{400}\right), 10^{-4}\right] & \text{if } z_b < 0 \end{cases}$$

where $c_f$ is a unitless coefficient, which in our study ranges from 0.1 (soft beds) to 1.0 (hard beds).

The parameters are chosen to simulate a difference between the observed ice volume of less than 1 mSLE and a deviation in the grounded area of less than 2%. This has to be fulfilled for the whole AIS. We found that these conditions matched a good PD state in terms of ice volume and extension.

To assess the grounding-line mismatch we plot the ice-mask difference between the simulated and the observed ice mask (1: grounded ice, 0: floating ice/ocean) for our parameter space that fulfilled the above condition. The lowest and best fit are then chosen as a quadratic sum of the mask difference (Figure 2). Overall we find more advanced EAIS and Ronne grounding lines. On the other hand, the Ross grounding line tends to retreat more than observations. These initial states fit well within the range of ISMIP6-Init (Seroussi et al., 2019), thus we believe that our approach can be considered as valid. This figure will be included also in the supplementary material of the manuscript.

[Figure]

*Figure 2: Lowest and best fit of our chosen ensemble parameters. Value zero represents the same mask value between the simulated and the observed ice mask. 1/-1 represent advanced/retreated grounded points with respect to the observations.*

**need to justify range choice for this parameter. 6 is awefully high**
**and the statement is misleading when you only have 3 adjustable parameters (going by table 1)**

The choice of this parameter was to cover the whole range that simulates realistic ice volumes and extensions, including extreme cases. It was based on Ma et al., (2010), who considered values up to 5 (though with lower enhancement factors for ice shelves). Actually, the value 6 did not simulate a realistic extension or ice volume. We will therefore remove it from the manuscript.

**Some additional constraints such as present-day basal temperature at EDC and EDML would offer some independent constraint on this parameter.**
*"Regarding ice dynamics, our analysis revealed that the enhancement factor has the strongest influence on the extension of ice"*

Figure 5 shows the PD mean simulated basal temperature of the AIS, highlighting the Dome C (EDC) and EDML ice-core locations. In the case of EDML all simulations reach the pressure-melting point, which leads to a temperate bed (Table 1). At Dome C we find that with the exception of two simulations, the rest yield a base temperature close to the pressure-melting point (the lowest temperature is -0.8 °C). These results are in agreement with observations and modeling studies (Van Liefferinge et al., 2018).

The simulated base temperature is therefore not only a function of the enhancement factor, but also dependent on other variables subject to uncertainty, such as geothermal heat flow, basal friction or water drainage system. Since the obtained values are similar, we do not believe that it can be used as a metric to constrain values of the enhancement factor.

[Figure]

*Figure 5: Mean simulated basal temperature highlighting the EDML and DomeC. Red color indicates a temperate base.*

```
Basal temperature [°C]
DomeC            EDML
* * *
-0.1             0.0
0.0              0.0
0.0              0.0
-0.5             0.0
-0.3             0.0
-0.3             0.0
-0.2             0.0
-0.2             0.0
-0.4             0.0
-0.2             0.0
-0.2             0.0
-0.2             0.0
-0.2             0.0
-0.2             0.0
-0.8             0.0
-0.7             0.0
-0.6             0.0
-0.6             0.0
-0.6             0.0
-0.5             0.0
-0.7             0.0
```

*Table 1: Simulated basal temperature for every simulation at DomeC and EDML.*

**References:**

- Ma, Ying, et al. "Enhancement factors for grounded ice and ice shelves inferred from an anisotropic ice-flow model." *Journal of Glaciology* 56.199 (2010): 805-812.
- Van Liefferinge, Brice, et al. "Promising Oldest Ice sites in East Antarctica based on thermodynamical modelling." *The Cryosphere* 12.8 (2018): 2773-2787.
- Seroussi, H., Nowicki, S., Simon, E., Abe-Ouchi, A., Albrecht, T., Brondex, J., Cornford, S., Dumas, C., Gillet-Chaulet, F., Goelzer, H., Golledge, N. R., Gregory, J. M., Greve, R., Hoffman, M. J., Humbert, A., Huybrechts, P., Kleiner, T., Larour, E., Leguy, G., Lipscomb, W. H., Lowry, D., Mengel, M., Morlighem, M., Pattyn, F., Payne, A. J., Pollard, D., Price, S. F., Quiquet, A., Reerink, T. J., Reese, R., Rodehacke, C. B., Schlegel, N.-J., Shepherd, A., Sun, S., Sutter, J., Van Breedam, J., van de Wal, R. S. W., Winkelmann, R., and Zhang, T.: initMIP-Antarctica: an ice sheet model initialization experiment of ISMIP6, *The Cryosphere*, 13, 1441–1471, https://doi.org/10.5194/tc-13-1441-2019, 2019.
- Coulon, V., Klose, A. K., Kittel, C., Edwards, T., Turner, F., Winkelmann, R., and Pattyn, F.: Disentangling the drivers of future Antarctic ice loss with a historically calibrated ice-sheet model, The Cryosphere, 18, 653–681, https://doi.org/10.5194/tc-18-653-2024, 2024.
- Seroussi et al. (*in rev.*): Evolution of the Antarctic Ice Sheet over the next three centuries from an ISMIP6 model ensemble.

---

## Referee Report (RR1)

**Review of Blasco et al.: "Antarctic Tipping points triggered by the mid-Pliocene warm climate"**

**By Tijn Berends**

The authors have put in a lot of effort to address the issues raised by myself and the other reviewer. In general, I think this has improved the manuscript, providing a better context for the modelling choices made by the authors in setting up their experiments, as well as for the conclusions they draw from their results.

The one point I think needs some more attention is the initialisation procedure. I don't entirely agree with the authors' rationale for not optimising their basal friction coefficients, and while I do not think this invalidates their experiments, I do think some more context is appropriate.

**Major points**

In your rebuttal, and in the newly updated Discussion section of your manuscript, you describe some additional experiments where you included a "nudging" procedure, adapting the spatially variable basal friction over time to achieve a stable ice thickness close to the present-day observations. You state that you performed this nudging over a period of 30,000 years, but that the ice-sheet "is not yet in equilibrium" at the end of this period. You illustrate this with the new Fig. S9, which shows the change in total ice volume over time in a 30,000 continuation run, where you keep the friction field fixed in its nudged state, and the climate fixed to the present-day. You state that similar problems were reported by Seroussi et al. (in review) and Coulon et al. (2023).

I do not believe Fig. S9 shows what you think it shows. During the first ~2,000 years, your modelled ice volume changes by about 1 m.s.l.e. I doubt that this is simply a continuation of the trend at the end of your 30,000-year nudging phase. Instead, this initial sharp change, followed by the slow relaxation you see after ~2,000 years, is indicative of a "model shock". This is a common problem when using a nudging approach; during the nudging phase, it is almost unavoidable to put some (temporary) restrictions on the modelled ice geometry, to prevent (parts of) the ice sheet from collapsing before the friction is sufficiently nudged to keep them stable. Some modellers artificially reduce the rates of thickness change during this phase, others limit how far the thickness is allowed to deviate from observed, yet others simply do not allow any thickness change at the grounding line. This is a tricky thing to get right, especially when you do not simultaneously nudge the sub-shelf melt rates (which you don't mention doing). While not knowing exactly what approach you chose, I suspect that the "jump" at the start of your continuation simulation is a result of problems from these implementations, rather than a fundamental problem with the nudging procedure (as there are several other models out there that have used it to achieve a much more stable ice sheet than what you have shown here).

Also, maybe a minor point, but I could not find any mention of such problems in the work by Coulon et al. (2023). Seroussi et al. (in review), as far as I'm aware, only mention these model drift problems for models that invert for velocities, rather than for geometry.

I think it's acceptable to show your results from the non-optimised simulations, but I don't think it's fair to dismiss the optimised simulations for the reasons you wrote. The argument that the nudging procedure (under a length list of assumptions) finds the present-day basal friction field, and that that might be different from the friction field during the Pliocene, is a better reason for not using it in this context. Also, I myself would have no problem with it if you simply stated that your optimisation approach is still a work in progress, and was not yet ready for application when you began your study.

**Minor points**

Abstract and other places: I am unfamiliar with the notation you use (e.g. …a mean contribution of $2.7^{+0.1}_{-0.4}$ mSLE to $7.0^{+0.1}_{-0.1}$ mSLE…). Are these uncertainties of uncertainties?

L 125-126 At the risk of offending Lev, I believe that the previous phrasing of "terrain-following coordinates" is more informative and easier to understand than "sigma-coordinates" (as not all models use the letter sigma for the scaled vertical coordinate). Perhaps "…terrain-following coordinates (also known as "sigma coordinates" in some models, e.g. …)…"?

L 141-142 "Here, cf is a dimensionless field representing the basal properties of the base, such as soft /hard beds (cf =0.1) or hard beds (cf =1.0)." This phrasing suggest that cf = cf(x,y), while in your rebuttal, you state the cf is "a unitless coefficient", which suggests that it has no spatial variability. Which one is it?

L 244 "…grounded-ice differs by less than 2% from observations…" Do you mean the grounded ice area or volume?

---

## Author Response (AR2)

Blue color:    Editor's/Reviewer comment
Black color:    Authors response

**=== Reviewer ===**

The authors have put in a lot of effort to address the issues raised by myself and the other reviewer. In general, I think this has improved the manuscript, providing a better context for the modelling choices made by the authors in setting up their experiments, as well as for the conclusions they draw from their results.

The one point I think needs some more attention is the initialisation procedure. I don't entirely agree with the authors' rationale for not optimising their basal friction coefficients, and while I do not think this invalidates their experiments, I do think some more context is appropriate.

We thank Tijn Berends for reviewing the manuscript again and for appreciating our effort. We have taken into account his major concerns in the new manuscript.

**Major points**

In your rebuttal, and in the newly updated Discussion section of your manuscript, you describe some additional experiments where you included a "nudging" procedure, adapting the spatially variable basal friction over time to achieve a stable ice thickness close to the present-day observations. You state that you performed this nudging over a period of 30,000 years, but that the ice-sheet "is not yet in equilibrium" at the end of this period. You illustrate this with the new Fig. S9, which shows the change in total ice volume over time in a 30,000 continuation run, where you keep the friction field fixed in its nudged state, and the climate fixed to the present-day. You state that similar problems were reported by Seroussi et al. (in review) and Coulon et al. (2023).

I do not believe Fig. S9 shows what you think it shows. During the first ~2,000 years, your modelled ice volume changes by about 1 m.s.l.e. I doubt that this is simply a continuation of the trend at the end of your 30,000-year nudging phase. Instead, this initial sharp change, followed by the slow relaxation you see after ~2,000 years, is indicative of a "model shock". This is a common problem when using a nudging approach; during the nudging phase, it is almost unavoidable to put some (temporary) restrictions on the modelled ice geometry, to prevent (parts of) the ice sheet from collapsing before the friction is sufficiently nudged to keep them stable. Some modellers artificially reduce the rates of thickness change during this phase, others limit how far the thickness is allowed to deviate from observed, yet others simply do not allow any thickness change at the grounding line.

This is a tricky thing to get right, especially when you do not simultaneously nudge the sub-shelf melt rates (which you don't mention doing). While not knowing exactly what approach you chose, I suspect that the "jump" at the start of your continuation simulation is a result of problems from these implementations, rather than a fundamental problem with the nudging procedure (as there are several other models out there that have used it to achieve a much more stable ice sheet than what you have shown here).

Also, maybe a minor point, but I could not find any mention of such problems in the work by Coulon et al. (2023). Seroussi et al. (in review), as far as I'm aware, only mention these model drift problems for models that invert for velocities, rather than for geometry.

I think it's acceptable to show your results from the non-optimised simulations, but I don't think it's fair to dismiss the optimised simulations for the reasons you wrote. The argument that the nudging procedure (under a length list of assumptions) finds the present-day basal friction field, and that that might be different from the friction field during the Pliocene, is a better reason for not using it in this context. Also, I myself would have no problem with it if you simply stated that your optimisation approach is still a work in progress, and was not yet ready for application when you began your study.

Indeed, in this case we did not nudge the sub-shelf melt rates, which could explain the drift jump observed in our simulations. Also, we did not allow for any temporal restrictions which could help for stability if we seek for a stable PD. Though we now have a nudging scheme with satisfactory results (Juarez-Martinez et al., 2024), by the time of this study it was still work under progress, therefore it is not completely implemented in our model version. We will remove Figures S8 and S9 showing results for optimized friction coefficients, since results are similar to non-optimized fields, and they should be analyzed with caution. We maintain our argument that the present-day basal friction field might be different from the friction field during the Pliocene.

Regarding the work of Coulon et al., (2023), they optimize towards PD ice thickness and simulate a trend as observed in Figure 2, under PD forcing. They simulate WAIS collapse by Year 3000. However, in their simulations they first nudge melt rates and later apply PD oceanic conditions which explains such a drift (personal communication).

**Minor points**
Abstract and other places: I am unfamiliar with the notation you use (e.g. …a mean contribution of $2.7^{+0.1}_{-0.4}$ mSLE to $7.0^{+0.1}_{-0.1}$ mSLE…). Are these uncertainties of uncertainties?

This uncertainty range refers to a particular AOGCM for dynamical parameters. We removed the uncertainties in the abstract since it can lead to confusion.

L 125-126 At the risk of offending Lev, I believe that the previous phrasing of "terrain-following coordinates" is more informative and easier to understand than "sigma-coordinates" (as not all models use the letter sigma for the scaled vertical coordinate). Perhaps "…terrain-following coordinates (also known as "sigma coordinates" in some models, e.g. …)…"?

We included both names in the description.

L 141-142 "Here, cf is a dimensionless field representing the basal properties of the base, such as soft /hard beds (cf =0.1) or hard beds (cf =1.0)." This phrasing suggest that cf = cf(x,y), while in your rebuttal, you state the cf is "a unitless coefficient", which suggests that it has no spatial variability. Which one is it?

Indeed, cf is no field. We changed it to value.

L 244 "…grounded-ice differs by less than 2% from observations…" Do you mean the grounded ice area or volume?

We mean area. We changed it in the description.

**References:**

- Juarez-Martinez, A., Blasco, J., Robinson, A., Montoya, M., and Alvarez-Solas, J.: Antarctic sensitivity to oceanic melting parameterizations, EGUsphere [preprint], https://doi.org/10.5194/egusphere-2023-2863, 2024.

**=== Editor ===**

Given the latest re-review by Tijn Berends and the current state of the revised thesis, to expedite completion, I'm requesting a few minor revisions, thereby avoiding a further external review stage. To make this work, please address Tijn's remaining concerns. Their main concern about the friction coefficients issue is I take mostly a communication issue of what your own tests imply. For even 100 kyr glacial cycle scales, I have strong concerns about PD optimization of friction coefficients as they generally do not account for changes in basal temperature over say a glacial cycle or within a surge/dormancy cycle of an ice stream.

Note however (contrary to Tijn's comment), "sigma coordinate" (if this is indeed what you are using) is a more precise (and standard, cf eg wikipedia..) term than terrain following coordinates (of which there are a number of different ones). So please retain "sigma coordinate" if accurate. If not accurate, do precisely specify the vertical coordinate.

We thank the Editor for considering minor revisions. We maintained the term sigma coordinates in the manuscript but added in parenthesis Tijn's suggestion of terrain-following coordinates.

Also, expanding on Tijn's comment about uncertainty of uncertainties, your interquartile range expressions are comprehensible in the main text given that you parenthetical assign the different values to different models, but for those who first read the abstract, this will be incomprehensible.

Indeed, since it can be confusing we removed the uncertainties in the abstract.

##################

**Concern 1:**
The classical central limit theorem (CLT) requires independent identically distribution (IID) random variables. What are your IID random variables?
The application of descriptive statistics does not equate to the specification of a probability distribution. And if 30 simulations from a high dimensional non-linear modelling space were all that was needed to make confident inferences, robust inference of past and future earth system change would be a lot easier than it is.
It is fine to use descriptive statistics to summarize ensemble results. Do make explicitly clear that your "probabilities" are empirical probabilities (ie relative frequencies) based on your ensemble results. Cf even the fine wikipedia page on this if you are not clear on the distinction between probability and empirical probability.
And please clarify on first use whether A \pm B (when taken from cited literature) is nominally one or two sigma ranges.

We agree that the distribution of sea-level equivalent with respect to the dynamic parameters does not follow a gaussian distribution, so the CLT should not be taken as valid in our study. We will make clear that our probabilities are empirical probabilities.

The uncertainty ranges of the cited literature refers to 1 sigma range. We made this clear in the following submission.

**Concern 2:**
The interpolation of AOGCM fields to the interior sub-ice-shelf regions is another potentially major source of uncertainty in your experiments, at least for the marine sectors of the AIS. Given the potential for changes in submarine melt driven grounding line retreat to induce the tipping points you are considering along with the potentially large discrepancy between an extrapolated AOGCM field to the actual temperature distribution under an ice shelf, this warrants at least a bit more consider discussion than the "slightly different final states" claim which I suspect is invalid. It's an ongoing a challenge for all of us modelling ice sheets with significant ice shelves.

We agree that interpolation of AOGCM fields in the ice-shelf cavities is a big uncertainty which we may have underestimated in our study. We reformulated as follows the following paragraph:

"Note that the AOGCMs do  not provide any oceanic information under Antarctic ice-shelf grid cells. Since that grid information is required to force our ice-sheet model, we interpolate to that grid point using the value of the nearest neighbor at the same depth. Of course, applying other interpolation schemes - and increasing the spatial resolution of the grid - will change the oceanic conditions and lead to different final states. The ideal outcome would be to include ocean models that resolve the ocean circulation below the shelves, which is not the case of the PlioMIP2 ensemble. Jourdain et al., (2020) propose an extrapolation protocol which is another possibility but it would add another source of uncertainty. We used the nearest neighbor interpolation scheme for simplicity but extrapolation of oceanic conditions inside ice-shelf cavities is an ongoing challenge within the scientific community."

**References:**

- Jourdain, N. C., Asay-Davis, X., Hattermann, T., Straneo, F., Seroussi, H., Little, C. M., and Nowicki, S.: A protocol for calculating basal melt rates in the ISMIP6 Antarctic ice sheet projections, The Cryosphere, 14, 3111–3134, https://doi.org/10.5194/tc-14-3111-2020, 2020.